# STRUCTURE LANGUAGE MODELS FOR PROTEIN CONFORMATION GENERATION

**Jiarui Lu**[1,2,†]**, Xiaoyin Chen**[1,2,†]**, Stephen Z. Lu**[1,3]**, Chence Shi**[1,2]**, Hongyu Guo**[4,5]**,
Yoshua Bengio**[1,2,6] **& Jian Tang**[1,6,7]

[1]Mila - Québec AI Institute, [2]Université de Montréal, [3]McGill University,
[4]University of Ottawa, [5]National Research Council Canada,
[6]CIFAR AI Chair, [7]HEC Montréal

## ABSTRACT

Proteins adopt multiple structural conformations to perform their diverse biological functions, and understanding these conformations is crucial for advancing drug discovery. Traditional physics-based simulation methods often struggle with sampling equilibrium conformations and are computationally expensive. Recently, deep generative models have shown promise in generating protein conformations as a more efficient alternative. However, these methods predominantly rely on the diffusion process within a 3D geometric space, which typically centers around the vicinity of metastable states and is often inefficient in terms of runtime. In this paper, we introduce Structure Language Modeling (SLM) as a novel framework for efficient protein conformation generation. Specifically, the protein structures are first encoded into a compact latent space using a discrete variational auto-encoder, followed by conditional language modeling that effectively captures sequence-specific conformation distributions. This enables a more efficient and interpretable exploration of diverse ensemble modes compared to existing methods. Based on this general framework, we instantiate SLM with various popular LM architectures as well as proposing the ESMDiff, a novel BERT-like structure language model fine-tuned from ESM3 with masked diffusion. We verify our approach in various scenarios, including the equilibrium dynamics of BPTI, conformational change pairs, and intrinsically disordered proteins. SLM provides a highly efficient solution, offering a 20-100x speedup than existing methods in generating diverse conformations, shedding light on promising avenues for future research.

## 1 INTRODUCTION

Protein structure dynamics are fundamental to understanding the biological functions of proteins. The ability of proteins to adopt multiple conformations is crucial for their function in influencing interactions with other biomolecules and the environment. Traditional computational methods, such as molecular dynamics (MD) simulations, have long been used to explore these dynamics. However, these methods are computationally expensive and time-consuming. Structure prediction models, such as AlphaFold 2 (Jumper et al., 2021) and RosettaFold (Baek et al., 2021), have made significant strides in predicting static protein structures, yet often fail to accurately capture the dynamic nature of proteins and their multiple conformations (Chakravarty & Porter, 2022).

Recently, significant progress has been made by adopting generative models to efficiently explore the complicated protein conformational space. For example, Noé et al. (2019) adopts normalizing flow to match the underlying Boltzmann distribution by learning from simulation data. Despite their potential, normalizing flow-based methods (Noé et al., 2019; Klein et al., 2023) face challenges in modeling large protein systems with hundreds of amino acids, as the invertibility constraint becomes a major obstacle when scaling up model parameters. As a remedy, denoising diffusion can efficiently learn from structural data (Jing et al., 2023; Lu et al., 2024b; Wang et al., 2024; Zheng et al., 2024), achieve good generalization, and perform amortized inference. However, modeling

---

[†]Equal contribution. Code available at `https://github.com/lujiarui/esmdiff`.
Correspondence to: `jiarui.lu@mila.quebec`, `jian.tang@hec.ca`.

high-dimensional protein structures explicitly in their 3D Euclidean space can demand intensive computation and usually requires accounting for special equivariant properties (Köhler et al., 2020). Furthermore, L2-based training objectives such as denoising score matching (Song et al., 2020) tend to predict local perturbations rather than capturing remote modes of alternative conformations (Wang et al., 2024). Consequently, these models may overallocate their capacity to learn structural noises in the training data instead of focusing on low-frequency structural changes (Chou, 1985).

In complement with existing approaches, we present Structure Language Modeling (SLM), a novel framework for protein conformation generation that performs generative modeling in the latent space of protein structures. Inspired by the recent progress in developing structural vocabularies for protein representation learning (Su et al., 2023; Hayes et al., 2024), our approach first encodes structural flexibility into a distribution over latent tokens using a discrete variational autoencoder, as illustrated in Fig. 1. The discrete latent encoding removes high-frequency details of protein structures, forming "structure languages" that effectively capture the uncertainty of complex protein conformations (Fig. 2a); Conditional language modeling is then applied to these latent structure tokens, using amino acid types as context to capture sequence-specific conformation distributions (Fig. 2b); Protein conformations can finally be reconstructed by mapping structure tokens into 3D space with a learned decoder (Fig. 2c). By leveraging generative language modeling in the discrete latent space, SLM bypasses the complexity of equivariant constraints associated with geometric symmetries and benefits from enhanced model capacity. As a general framework, SLM is fully compatible with any existing language model (LM) architectures and shows promising scalability. To further demonstrate the versatility of our approach, we introduce ESMDiff, a novel BERT-like structure language model instantiation fine-tuned from ESM3 (Hayes et al., 2024) with masked discrete diffusion (Austin et al., 2021; Zhao et al., 2024) grounded in the SLM framework. Experimental results across various conformation generation scenarios demonstrate the state-of-the-art performance of SLM including the representative ESMDiff model, achieving orders of magnitude faster speeds compared to existing generative methods. The proposed framework paves the way for new research avenues in addressing the protein conformation sampling challenge.

We summarize our key contributions as follows.

- We comprehensively explore an innovative conformation generation framework based on language modeling in the latent space, which opens up potential research avenues.

- We introduce ESMDiff, a novel fine-tuned variant of a state-of-the-art protein language model, built on masked discrete diffusion.

- We demonstrate the superior capability of structure language models by evaluating them on various conformation generation settings and comparing them with existing methods.

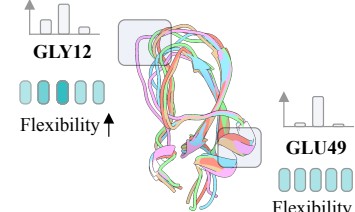

Figure 1: Residue flexibility (BPTI clusters, Shaw et al. (2010)) reflected by the categorical distribution over latent structure tokens. Different tokens (colored in different shades ) are used to encode varying local structural patterns.

## 2 RELATED WORK

**Protein language models.** In recent years, several language models of protein sequence have been built. Among these, the ESM-series (Rives et al., 2021; Lin et al., 2023; Hayes et al., 2024) and other similar models (Elnaggar et al., 2021; Alley et al., 2019) have garnered great attention because of their wide range of downstream applications such as protein engineering (Meier et al., 2021). On the other hand, auto-regressive protein language models, based on either recurrent neural networks (Alley et al., 2019), or Transformer including ProGen (Madani et al., 2023) and ProtGPT2 (Ferruz et al., 2022), are able to generate *de novo* sequences with input controlling tokens. Specially, inverse folding models (Ingraham et al., 2019; Jing et al., 2020; Hsu et al., 2022; Dauparas et al., 2022; Gao et al., 2022) learn to perform structure-based protein design with geometric-aware encoders.

**Generative conformation sampling.** Given the intensive computation of traditional MD simulations, generative models have been used to learn conformation distributions in a data-driven fashion. The Boltzmann generator (Noé et al., 2019) uses normalizing flow to fit the Boltzmann distribution from target-specific simulation data. Arts et al. (2023) extends this by using denoising

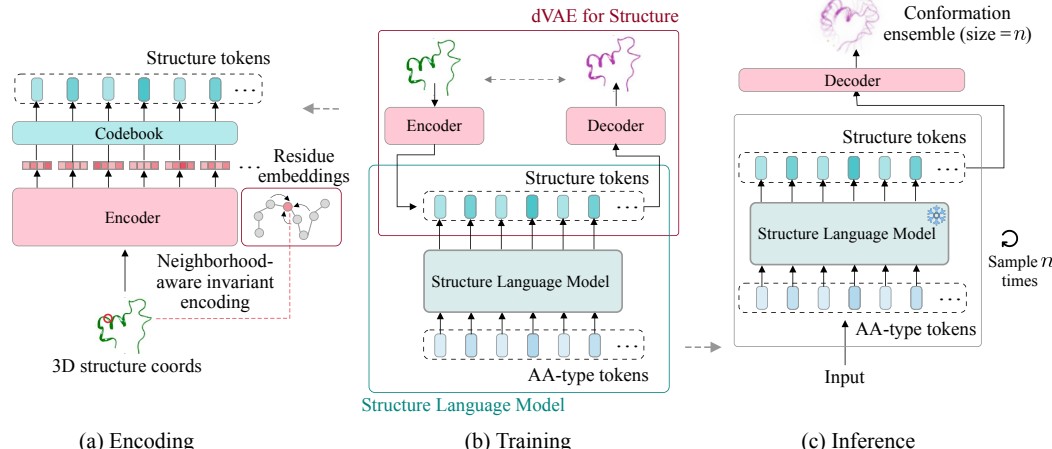

Figure 2: An illustration of the proposed SLM framework.

diffusion models for coarse-grained protein conformations. Furthermore, EigenFold (Jing et al., 2023), Str2Str (Lu et al., 2024b), AlphaFlow (Jing et al., 2024a), ConfDiff (Wang et al., 2024), and DiG (Zheng et al., 2024) leverage diffusion or flow matching to conditionally sample protein conformations by learning from PDB data. Recently, AlphaFold3 (Abramson et al., 2024) revised the structure decoder of AlphaFold2 to a diffusion-based module for diversified structure prediction.

**Quantized representation for protein structures.** Beside the prevailing diffusion models for protein structure, representation learning of protein structures using discrete variational autoencoders (dVAE) has gained increasing attention in recent years. FoldSeek (van Kempen et al., 2022) is one of the earliest attempt to build dVAE for fast structure search and alignment. Based on this, SaProt (Su et al., 2023) constructs learned representations with both sequence and structure tokens as input, while ProtT5 (Heinzinger et al., 2023) fine-tuned an existing language model to accept structure tokens as input. PVQD (Haiyan et al., 2023) applied latent diffusion in the embedding space of dVAE for conditional protein structure generation. ProSST (Li et al., 2024) trained an autoencoder with K-means clustering applied in the latent space. Gaujac et al. (2024) and Gao et al. (2024) respectively build dVAE with large vocabularies for learning protein structure representations.

***Remarks:*** Our work is closely related to these concurrent research directions by leveraging LMs to model and efficiently perform conformation generation over the quantized representation of protein structures. We refer to this framework as "structure language models" and describe them in detail.

## 3    PROTEIN CONFORMATION GENERATION WITH LANGUAGE MODELING

**Notation.** A protein with $L$ residues is identified by its sequence of amino acid types $c \in |\mathcal{S}|^L$ where $\mathcal{S}$ is the vocabulary of 20 standard amino acids. The protein (backbone) structure is represented by its composing 3D atom positions $x \in \mathcal{X} \equiv \mathbb{R}^{L \times 4 \times 3}$ including all backbone heavy atoms.

### 3.1    LEARNING THE SEQUENCE-STRUCTURE DISTRIBUTION

To address the conformation generation problem, we start with modeling the sequence-to-structure translation distribution $p(x|c)$ of interest and derive the learning objective in this section. To circumvent explicitly learning in the structure space, the roto-translation *invariant*[*] latent representation $z$ is introduced to encode 3D atomic protein structure, where $z \equiv (z_1, \ldots, z_L) \in |V|^L$. Given this, the target distribution $p(x|c)$ can be derived by marginalizing the joint distribution $p(x|c) = \int_z p(x, z|c)$. We further factorize this joint distribution according to the *Bayes' rule* by isolating the latent variable $z$: $p_{\theta,\phi}(x, z|c) = p_\phi(x|c, z)p_\theta(z|c)$, where $p_\phi$ denotes the (de-

---

[*]For example, features like distance and angle are roto-translation invariant. This relationship can be formally written as $q(z|T \circ x) \triangleq q(z|R \circ x + t) = q(z|x), \forall T$.

coding) distribution over the 3D protein structures given the structure token and sequence, and $p_\theta$ denotes the conditional distribution over the structure tokens, respectively modeled by neural networks with parameter set $\phi, \theta$. This gives rise to the evidence lower bound on the likelihood of model distribution over protein structures conditioned on sequence:

$$\log p_{\theta,\psi}(\boldsymbol{x}|\boldsymbol{c}) \geq \mathbb{E}_{q_\psi(\boldsymbol{z}|\boldsymbol{x})}\left[\log p_\phi(\boldsymbol{x}|\boldsymbol{c}, \boldsymbol{z})\right] - D_{\mathrm{KL}}(q_\psi(\boldsymbol{z}|\boldsymbol{x})\|p_\theta(\boldsymbol{z}|\boldsymbol{c})) \triangleq \mathcal{L}(\phi, \theta), \qquad (1)$$

where $\psi$ is introduced to parameterized the posterior distribution over latent representation $\boldsymbol{z}$. Please refer to the Appendix G.2.1 for the full derivation of Eq. (1). Directly optimizing the right-hand side of Eq. (1) can be intractable and difficult since we have unknown posterior $q_\psi$. In practice, one may adopt an one-step expectation–maximization (EM) approach (Dempster et al., 1977) by first jointly learning $p_\phi$ and $q_\psi$ with a simple and parameter-free prior distribution $p(\boldsymbol{z}|\boldsymbol{c})$, followed by optimization on $p_\theta$ with the learned $p_\phi^*$ and $q_\psi^*$, similar to Van Den Oord et al. (2017). This yields the overall two-stage and separable training pipeline:

**I. Learning quantized representation for structure.** With the prior $p(\boldsymbol{z}|\boldsymbol{c})$ fixed, we begin by maximizing the ELBO $\mathcal{L}(\phi, \theta)$ with respect to the encoder $\psi$ and decoder $\phi$, using protein structure samples $\mathcal{D} = \{(\boldsymbol{c}, \boldsymbol{x})\}$. In the context of discrete latent spaces, this process is analogous to training a discrete VAE (dVAE) (Van Den Oord et al., 2017) to learn quantized representations for protein structures. Here, the encoder $q_\psi(\boldsymbol{z}|\boldsymbol{x})$ maps structures to latent tokens, while the decoder $p_\phi(\boldsymbol{x}|\boldsymbol{z}, \boldsymbol{c})$ reconstructs structures from these tokens[†]. The prior $p(\boldsymbol{z}|\boldsymbol{c})$ is fixed to be uniform during this stage.

**II. Learning the prior over latent tokens.** In this stage, we fix the learned parameters $\phi^*$ and $\psi^*$, and train the prior $p_\theta$ by maximizing the ELBO: $\arg\max_\theta \mathcal{L}(\phi^*, \theta)$. Since both $\phi^*$ and $\psi^*$ are fixed, the reconstruction term in the ELBO cancels out, and training reduces to minimizing the KL divergence $D_{\mathrm{KL}}(q_{\psi^*}\|p_\theta)$. This is equivalent to performing maximum likelihood estimation, as $\mathbb{E}_{(\boldsymbol{c},\boldsymbol{x})\sim\mathcal{D}}\mathbb{E}_{\boldsymbol{z}\sim q_\psi(\boldsymbol{z}|\boldsymbol{x})}p_\theta(\boldsymbol{z}|\boldsymbol{c})$ with respect to $p_\theta$. Given that both $\boldsymbol{z}$ and $\boldsymbol{c}$ are categorical variables, this formulation resembles a *translation* task, allowing $p_\theta$ to be parameterized by language models.

Optimizing Eq. (1) provides a general learning framework for conformation generation. In practice, we can approach this objective in a demystified view: we are exploring new "conformations" in an invariant latent space via learning a sequence-to-structure (*seq2str*) network while offloading the complicated geometric modeling to the structure auto-encoder. This allows the practitioners to choose from popular architectures of structure encoders/decoders and modern language models.

## 3.2 STRUCTURE LANGUAGE MODELING

The prior learned in the previous stage is now applied to conformation generation, which can be framed as a conditional generative modeling problem for the *seq2str* translation. Given an input condition $\boldsymbol{c} \in |\mathcal{S}|^L$ which determines the molecular topology, the goal is to sample a conformation ensemble from $p(\boldsymbol{x}|\boldsymbol{c})$. To do this, we first sample a set of latent variables from the prior distribution learned earlier, $\boldsymbol{z} \sim p_\theta(\boldsymbol{z}|\boldsymbol{c})$, and then decode these latents using the decoder $p_\phi(\boldsymbol{x}|\boldsymbol{c}, \boldsymbol{z})$. The decoder is jointly trained with the encoder $q_\psi(\boldsymbol{z}|\boldsymbol{x})$ in the first stage, ensuring that the sampled latents align with the reconstruction. This framework supports roto-translation invariant inference and is described in Algorithm 1. Next, we illustrate this approach with two straightforward examples of structure language models (SLM): the encoder-decoder and decoder-only architectures.

**Encoder-decoder.** Given the conditional nature of translation, the prior $p_\theta(\boldsymbol{z}|\boldsymbol{c})$ can be explicitly modeled by an encoder-decoder architecture like T5 (Raffel et al., 2020). The decoder conditions on the context $\boldsymbol{c}$ and factorizes the structure tokens sequentially: $p(\boldsymbol{z}|\boldsymbol{c}) = \prod_{l=1}^L p(\boldsymbol{z}_l|\boldsymbol{z}_{<l}, \boldsymbol{c})$, where $\boldsymbol{z} \in \mathcal{Z}$ represents the quantized structure tokens. The training objective is the negative log-likelihood (NLL) loss conditioned on $\boldsymbol{c}$: $\mathcal{L}(\theta) = -\mathbb{E}_{(\boldsymbol{c},\boldsymbol{x})\sim\mathcal{D}}\mathbb{E}_{\boldsymbol{z}\sim q(\boldsymbol{z}|\boldsymbol{x})}\sum_{l=1}^L \log p_\theta(\boldsymbol{z}_l|\boldsymbol{z}_{<l}, \boldsymbol{c}), \boldsymbol{z}_{<1} \equiv \varnothing$.

**Decoder-only.** Alternatively, the latent prior $p_\theta(\boldsymbol{z}|\boldsymbol{c}) \propto p_\theta(\boldsymbol{c}, \boldsymbol{z})$ can be modeled autoregressively using a decoder-only architecture, such as GPT (Radford et al., 2019), where $\boldsymbol{c}$ serves as the "prompt". We define $\boldsymbol{y} \triangleq [\boldsymbol{c}, \boldsymbol{z}] = [\boldsymbol{c}^1, \dots, \boldsymbol{c}^L, \boldsymbol{z}^1, \dots, \boldsymbol{z}^L]$, and the training involves maximizing the likelihood over $\boldsymbol{y}$ via the NLL minimization: $\mathcal{L}(\theta) = -\mathbb{E}_{(\boldsymbol{c},\boldsymbol{x})\sim\mathcal{D}}\mathbb{E}_{\boldsymbol{z}\sim q(\boldsymbol{z}|\boldsymbol{x})}\sum_{l=1}^{2L} \log p_\theta(\boldsymbol{y}_l|\boldsymbol{y}_{<l})$, where $\boldsymbol{c}, \boldsymbol{x} \sim \mathcal{D}$ is the *i.i.d.* samples from the data dis-

---

[†]We assume that $\boldsymbol{x}$ is conditionally independent of $\boldsymbol{c}$ given latent variable $\boldsymbol{z}$ for simplicity. In practice, this leads to structure decoder $p_\phi(\boldsymbol{x}|\boldsymbol{z}, \boldsymbol{c}) \approx p_\phi(\boldsymbol{x}|\boldsymbol{z})$ such as in Hayes et al. (2024).

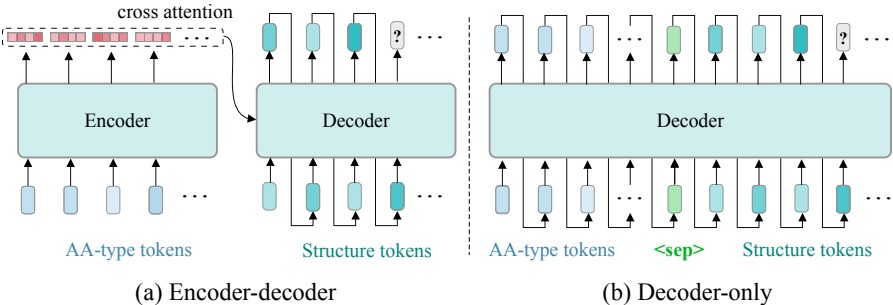

Figure 3: Autoregressive prior modeling the for latent structure tokens discussed in Section 3.2.

tribution over structures each with the associated amino-acid sequence as condition $c$. In practice, we add an additional special token [sep] to differentiate between these two modalities.

**Inference** involves sampling with a left-to-right decoding order, as defined by the autoregressive factorization of both language models. Fig. 3 briefly illustrates these two modeling strategies.

## 4 ESMDIFF: A MASKED DIFFUSION INSTANTIATION

Building on the foundation of SLM, we here propose ESMDiff as an instantiation based on discrete diffusion models (Austin et al., 2021). ESMDiff incorporates the inductive bias of *seq2str* translation and leverages the protein foundation model ESM3 (Hayes et al., 2024) through masked diffusion fine-tuning. The effectively fine-tuning of ESMDiff also exemplifies how a large pretrained BERT-like masked language model can be adapted to acquire conditional generative capabilities, making it well-suited for broader downstream tasks such as conformation generation.

### 4.1 CONDITIONAL MASKED DIFFUSION LANGUAGE MODEL

The discrete diffusion models can be generally defined by a sequential process of progressive noisy discrete variables $z_t \in V$ from the categorical variable $z_0 \in V$.[‡] Masked diffusion (Austin et al., 2021; Lou et al., 2023; Shi et al., 2024; Sahoo et al., 2024) represents a special case in which the transition includes an "absorbing state", denoted as [MASK]. In this formulation, the stationary distribution assigns all probability mass to the unique special token [MASK], such that $P(z = [MASK]) = 1$ and $P(z \neq [MASK]) = 0$. For convenience, we define $\boldsymbol{p}_M \in \{0,1\}^{|\bar{V}|}$ ($\bar{V} \triangleq V \cup \{[MASK]\}$) as the one-hot vector representing [MASK]. In masked diffusion, the stochastic forward process maps $z_0 \rightarrow [MASK]$ and remains in this state thereafter (i.e., "absorbing"). Conversely, the reverse process gradually unmasks (denoises) the [MASK] token to produce the data sample $z_0$, where $s < t$ (see Appendix G.2.2 for derivation):

$$q(\boldsymbol{z}_s|\boldsymbol{z}_t, \boldsymbol{z}_0) = \text{Cat}\left(\boldsymbol{z}_s; [\beta(s,t) + (1 - \lambda_M(\boldsymbol{z}_t))(1 - \beta(s,t))]\boldsymbol{z}_t + \lambda_M(\boldsymbol{z}_t)(1 - \beta(s,t))\boldsymbol{z}_0\right), \quad (2)$$

where $\beta(s,t) = \frac{1-\alpha(s)}{1-\alpha(t)}$ and $\lambda_M(\boldsymbol{z}_t) = \langle \boldsymbol{p}_M, \boldsymbol{z}_t \rangle$. Eq. (2) implies when $\boldsymbol{z}_t \neq [MASK]$, the backward process simply copies the unmasked token by $\boldsymbol{z}_s \leftarrow \boldsymbol{z}_t$, i.e. $q(\boldsymbol{z}_s|\boldsymbol{z}_t, \boldsymbol{z}_0) = \text{Cat}(\boldsymbol{z}_s; \boldsymbol{z}_t)$; otherwise the probability mass interpolates between $\boldsymbol{p}_M$ and $\boldsymbol{z}_0$. The posterior $q(\boldsymbol{z}_s|\boldsymbol{z}_t, \boldsymbol{z}_0)$ can be approximated by $p_\theta(\boldsymbol{z}_s|\boldsymbol{z}_t)$ using re-parameterization: $p_\theta(\boldsymbol{z}_s|\boldsymbol{z}_t) = q(\boldsymbol{z}_s|\boldsymbol{z}_t, \boldsymbol{u}_\theta(t, \boldsymbol{z}_t))$, where the neural net $\boldsymbol{u}_\theta \in \Delta^{|\bar{V}|}$ is a neural network that outputs a probability vector that remains in $\Delta^{|\bar{V}|}$. Unlike open-ended text generation, protein conformation generation is well-defined within the discrete diffusion models, as it conditions on the input amino acid sequence, allowing each output token to correspond uniquely to a position in the input and thus enjoy a fixed-length context window [§].

For conformation generation, we now consider the conditional case of masked diffusion. Given the amino acid types $c$, our goal is to sample structure tokens through Eq. (2), utilizing a conditional posterior $q(\boldsymbol{z}_s|\boldsymbol{z}_t, \boldsymbol{z}_0; \boldsymbol{c})$. This posterior can be re-parameterized similarly by incorporating the condition into the backbone model, resulting in $p_\theta(\boldsymbol{z}_s|\boldsymbol{z}_t; \boldsymbol{c}) = q(\boldsymbol{z}_s|\boldsymbol{z}_t, \boldsymbol{u}_\theta(t, \boldsymbol{z}_t, \boldsymbol{c}))$. To achieve this

---

[‡]See Appendix G.1 for more details of the discrete diffusion models.

[§]This indicates that $\boldsymbol{c}_{[i]}$ and $\boldsymbol{z}_{[i]}$ are aligned at the same position index $i$.

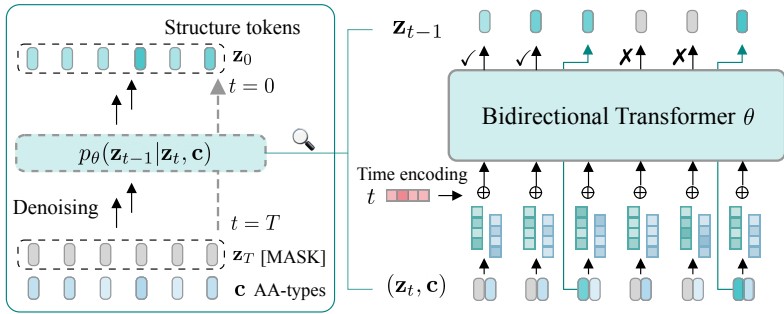

Figure 4: Illustration for conditional denoising network $\boldsymbol{u}_\theta(\boldsymbol{z}_t, \boldsymbol{c})$ where the masked (structure) tokens are colored grey. The unmasked tokens are carried over to the output without update, while the masked tokens are under random transition into either unmasked (✓) or still masked (✗) state.

goal, the reverse process simulated by $p_\theta(\boldsymbol{z}_s|\boldsymbol{z}_t; \boldsymbol{c})$ must effectively approximate the data distribution $p(\boldsymbol{z}|\boldsymbol{c})$. A feasible training objective is to optimize the estimation of the conditional ELBO within the continuous-time integral, resulting in the following loss (see Appendix G.2.4 for details):

$$\mathcal{L}(\theta) = \mathbb{E}_{\boldsymbol{c}, \boldsymbol{z}_0} \left\{ \int_{t \in [0,1)} \mathbb{E}_{\boldsymbol{z}_t \sim q(\boldsymbol{z}_t|\boldsymbol{z}_0)} \left[ \frac{1}{1-\alpha(t)} \frac{\partial \alpha(t)}{\partial t} \lambda_M(\boldsymbol{z}_t) \log\langle \boldsymbol{u}_\theta(t, \boldsymbol{z}_t, \boldsymbol{c}), \boldsymbol{z}_0 \rangle \right] \mathrm{dt} \right\}, \quad (3)$$

where $\boldsymbol{z}_0$ is sampled from the learned encoder $q_\phi(\boldsymbol{z}|\boldsymbol{x})$ with the corresponding amino acid condition $\boldsymbol{c}$ from the data distribution $p(\boldsymbol{x}, \boldsymbol{c})$, and $\lambda_M(\boldsymbol{z}_t)$ implies the loss is only applied for the latents $\forall t$, s.t. $\boldsymbol{z}_t = [\text{MASK}]$. In practice, we can employ Monte Carlo estimation to compute the integral.

---

**Algorithm 1** Inference: Conformation Generation of SLM

---

1: **Require:** amino acid types (condition) $\boldsymbol{c}$, generic conditional language models $p_\theta(\boldsymbol{z}|\boldsymbol{c})$, structure decoder $p_\phi(\boldsymbol{x}|\boldsymbol{c}, \boldsymbol{z})$, sampling temperature $T > 0$, sample size $N > 0$.
2: Initialize an empty set of samples $X = \varnothing$
3: **for** $i = 1$ to $N$ **do**
4:     $\boldsymbol{z}^{(i)} \sim p_\theta(\boldsymbol{z}|\boldsymbol{c})$                    ▷ Sample latent with temperature $T$
5:     $\boldsymbol{x}^{(i)} \sim p_\phi(\boldsymbol{x}|\boldsymbol{c}, \boldsymbol{z}^{(i)})$                    ▷ Decode the sampled latent vector
6:     $X \leftarrow X \cup \{\boldsymbol{x}^{(i)}\}$                    ▷ Add the decoded structure
7: **return** $X$

---

### 4.2 BIDIRECTIONAL ENCODER AS DENOISING NETWORK

We now discuss the implementation of the conditional denoising network using bidirectional encoder language models, such as BERT (Devlin, 2018). First, consider the sequential generalization of masked diffusion with a sequence of categorical variables. Let $\boldsymbol{z}_t$ now be a sequence of discrete structure tokens $[\boldsymbol{z}_{t,[1]}, \boldsymbol{z}_{t,[2]}, \ldots, \boldsymbol{z}_{t,[L]}]$ where $\boldsymbol{z}_{t,[i]} \in \bar{V}, \forall i = 1, \ldots, L$. Due to the interpolation scheme of Eq. (2), we assume conditional independence and factorize the posterior distribution $p_\theta(\boldsymbol{z}_s|\boldsymbol{z}_t; \boldsymbol{c})$ across the $L$ output tokens, such that $p_\theta(\boldsymbol{z}_s|\boldsymbol{z}_t, \boldsymbol{c}) = \prod_{i=1}^{L} p_\theta(\boldsymbol{z}_{s,[i]}|\boldsymbol{z}_t, \boldsymbol{c}) = \prod_{i=1}^{L} q\left(\boldsymbol{z}_{s,[i]}|\boldsymbol{z}_{t,[i]}, \boldsymbol{u}_{\theta,[i]}(t, \boldsymbol{z}_t, \boldsymbol{c})\right)$ where $\boldsymbol{u}_{\theta,[i]}(t, \boldsymbol{z}_t, \boldsymbol{c})$ represents the $i$-th output channel of neural network. This implement coincides with the BERT-style transformer architecture and allow us to take advantage of existing protein foundation model, for example ESM3 (Hayes et al., 2024). For a sequence of tokens, the masked log-term in the training objective from Eq. (3) is replaced by the summation: $\sum_{i=1}^{L} \lambda_M(\boldsymbol{z}_{t,[i]}) \log\langle \boldsymbol{u}_{\theta,[i]}(t, \boldsymbol{z}_t, \boldsymbol{c}), \boldsymbol{z}_{0,[i]} \rangle$, with notations the same as defined above.

**Modifications.** The following are special considerations for the network: (1) *Position-coupled encoding.* Unlike general translation problem, SLMs maintain strict position-to-position correspondence between amino acid types and the latent tokens[¶]. This inductive bias enables us to construct

---

[¶] Underlying co-evolutionary relationships between residues are shared across both amino acid types and its spatial patterns.

the input embedding for all position $i$ as follows: $\boldsymbol{e}_{[i]} = f_\theta[e_z(\boldsymbol{z}_{t,[i]}) + e_c(\boldsymbol{c}_{[i]}) + e_t(t)] \in \mathbb{R}^D$, where $e_z : |\bar{V}| \mapsto \mathbb{R}^D, e_c : |\mathcal{S}| \mapsto \mathbb{R}^D, e_t : \mathbb{R} \mapsto \mathbb{R}^D$ are the embedding functions and $f_\theta$ is a linear transformation. (2) *Copying*. The unmasked tokens $\boldsymbol{z}_{t,[i]} \neq$ [MASK] remain the same in spite of the model output. (3) *Zero-out* [MASK] . Since $\boldsymbol{u}_\theta$ parameterize the approximated clean data $\boldsymbol{z}_0$ (fully unmasked), the [MASK] token cannot present in the output and its probability should be zero-out. This is equivalent to adding $-\infty$ to the logit. In our study, the pre-trained LM head of ESM3 is replaced with a randomly initialized head with augmented vocabulary ($\bar{V}$) during fine-tuning.

## 5 EXPERIMENTS

**Base settings.** We start with the pre-trained dVAE established in Hayes et al. (2024) as the structure tokenizer (frozen). The structure quantization is perform residue-level with a receptive field over local geometric neighborhoods of protein structure. The structure language models as described in Section 3.2 are based off state-of-the-art language models as follows: (1) *S-T5* (384M) adopts the T5 (Raffel et al., 2020) architecture with a bidirectional encoder and an autoregressive decoder. (2) *S-GPT* (961M) models the joint distribution with an uni-directional decoder like GPT2 (Radford et al., 2019). (3) *ESM3 (zero shot)* is a pre-trained BERT model over multi-modal data and perform zero-shot inference by the iterative decoding. (4) *ESMDiff* is a fine-tuned variant of (3) on the PDB data using the masked diffusion objective in Eq. (3). For S-T5 and S-GPT, we embed the sequence tokens with the pre-trained ESM3-1.4B encoder to provide model condition. For ESMDiff, two different paradigms: Iterative Decoding (ID) and DDPM are considered. See Appendix B for details.

Table 1: Evaluation results of different conformation generation methods on generating the BPTI conformations. *JS-\** represents the Jensen-Shannon (JS) divergence between the sampled ensemble and long MD ensemble on pairwise distance (PwD), time-lagged independent components (TIC), and radius of gyration (Rg). The validity as the frequency of clash-free samples in the ensemble is also included. Moreover, the ensemble TM-score and ensemble RMSD are reported with respect to the five kinetic clusters in Shaw et al. (2010). The best results among all methods are **bold** while the best results among the SLM family are underlined (similarly applied to all tables below).

| | Method | JS-PwD ($\downarrow$) | JS-TIC ($\downarrow$) | JS-RG ($\downarrow$) | Validity ($\uparrow$) | TM-ens ($\uparrow$) | RMSD-ens ($\downarrow$) |
|---|---|---|---|---|---|---|---|
| MSA-based | MSA-Subs. | 0.593 | 0.482 | 0.742 | 0.990 | 0.840 | 1.526 |
| | AlphaFlow | 0.503 | 0.462 | 0.777 | **1.000** | **0.845** | 1.441 |
| Seq-based | EigenFold | 0.536 | 0.466 | 0.824 | 0.620 | 0.840 | 1.473 |
| | Str2Str (PF) | 0.561 | 0.590 | 0.325 | **1.000** | 0.705 | 2.155 |
| | Str2Str (SDE) | 0.506 | 0.552 | **0.302** | 0.960 | 0.664 | 2.480 |
| | ESMFlow | 0.524 | 0.439 | 0.804 | 0.970 | 0.839 | 1.462 |
| SLM | S-T5 | 0.410 | 0.446 | 0.528 | 0.740 | **0.845** | 1.434 |
| | S-GPT | 0.573 | 0.687 | 0.415 | 0.750 | 0.841 | 1.476 |
| | ESM3 (zero shot) | 0.406 | 0.445 | 0.561 | 0.800 | 0.842 | 1.450 |
| | ESMDiff (ID) | 0.422 | 0.432 | 0.510 | 0.910 | 0.844 | **1.421** |
| | ESMDiff (DDPM) | **0.372** | **0.420** | 0.439 | 0.940 | 0.843 | 1.437 |

**Baselines.** We consider multiple open-source models as evaluation baselines for the protein multiple conformation generation, which are mainly categorized into (1) MSA-based methods that includes *MSA-Subsampling* (Del Alamo et al., 2022; Bryant & Noé, 2024) and *AlphaFlow* (Jing et al., 2024a). These methods rely on inference-time retrieval of multiple sequence alignments (MSA) ; (2) Single sequence-based methods: *EigenFold* (Jing et al., 2023) leverages a harmonic diffusion process conditioned on OmegaFold (Wu et al., 2022) embeddings to generate protein structures, *Str2Str* (Lu et al., 2024b) simulates a round-trip local diffusion conditioned on input structure to explore hypothetical conformations, and ESMFlow (Jing et al., 2024a) replaces AlphaFlow with ESMFold as the backbone; (3) Specially tailored for intrinsically disordered proteins (IDPs) generation of idp-GAN (Janson et al., 2023). Results reported for baselines are obtained by re-running the inference pipeline and based on their open-source codes. The detailed pipeline can be found in Appendix A.

**Training data.** The training data for structure language models are controlled to contain only PDB entries on or before May 1st, 2020. This cutoff is aligned with previous works (Jing et al., 2023; 2024a; Lu et al., 2024b; Hayes et al., 2024) trained on PDB data to make fair comparison. The training set is further filtered to include all monomeric structures with a max resolution of 5.0Å, length ranging from 10 to 1000, which forms a total size of $|\mathcal{D}| = 112.4k$ as the training data.

**Participating benchmark sets.** To discover the potential of SLMs in conformation sampling, several relevant datasets are considered for benchmarking purpose: (1) simulation dynamics of BPTI (Shaw et al., 2010), (2) conformational changing pairs including the fold-switching (Chakravarty & Porter, 2022) and ligand-induced apo/holo states (Saldaño et al., 2022), and (3) intrinsically disordered proteins (IDPs) deposited in the protein ensemble database (PED) (Lazar et al., 2021) . These benchmarking tasks reflect different characteristics and challenges of the conformation generations, which provides a comprehensive evaluation for models.

## 5.1 Structural dynamics of BPTI

In the first experiment, we evaluate by generating the conformations of protein bovine pancreatic trypsin inhibitor (BPTI). The structural dynamic patterns of BPTI are well acknowledged in Shaw et al. (2010) with 1ms-long MD simulations, based on which five kinetic clusters have been revealed. Similar to Lu et al. (2024b), we report the Jensen Shannon (JS) divergence for distributions of *pairwise distance* (PwD), *time-lagged independent components* (TIC), and *radius of gyration* (Rg); the clash-free validity and the ensemble TM-score and root-mean-square-deviation (RMSD) w.r.t. the kinetic clusters. The benchmark results are shown in Table 1. Following Wang et al. (2024), we also evaluate the best distance of the generated samples to each cluster, as shown in Table 2. The RMSD and TM-score are both calculated using the TM-score binary (Zhang & Skolnick, 2004) with structural alignment. Note that the Cluster 3 is a difficult remote folding mode (Wang et al., 2024), yet SLMs achieve a significant improvement by modeling with a smaller matching RMSD.

Table 2: Evaluating on the best matching RMSD with respect to each of the kinetic clusters reported in Shaw et al. (2010) with an non-exhaustive ensemble size of $N = 100$. $\text{RMSD}_{\text{Ci}}(i = 1, 2, 3, 4, 5)$ are the lower the better ($\downarrow$). Among them, the Cluster 3 is the most challenging case to model (Wang et al., 2024) and highlighted in red , of which the best sample from ESMDiff is visualized.

| | Method | RMSD$_{C1}$ | RMSD$_{C2}$ | RMSD$_{C3}$ | RMSD$_{C4}$ | RMSD$_{C5}$ |
|---|---|---|---|---|---|---|
| MSA-based | MSA-Subs. | 0.953 | 1.669 | 2.412 | 1.616 | 0.982 |
| | AlphaFlow | 0.882 | 1.693 | 2.418 | 1.380 | **0.915** |
| Seq-based | EigenFold | 0.905 | 1.680 | 2.478 | **1.352** | 0.977 |
| | Str2Str (PF) | 1.968 | 1.881 | 2.480 | 2.064 | 1.900 |
| | Str2Str (SDE) | 2.334 | 2.295 | 2.924 | 2.271 | 2.240 |
| | ESMFlow | 0.883 | 1.720 | 2.385 | 1.361 | 0.960 |
| SLM | S-T5 | **0.863** | 1.628 | 2.285 | 1.428 | 0.968 |
| | S-GPT | 0.922 | 1.687 | 2.357 | 1.462 | 0.953 |
| | ESM3 (zero shot) | 0.968 | **1.560** | 2.301 | 1.444 | 0.977 |
| | ESMDiff (ID) | 0.872 | 1.620 | 2.270 | 1.415 | 0.927 |
| | ESMDiff (DDPM) | 0.924 | 1.628 | **2.198** | 1.435 | 1.000 |

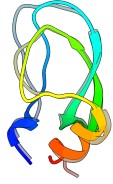

RMSD : 2.198 Å

Cluster 3

Table 3: Evaluation on the conformation changing pairs (Saldaño et al., 2022; Chakravarty & Porter, 2022): (1) Pearson correlation $r$ between sampled diversity and ground-truth diversity measured by the residue flexibility (*ResFlex*, absolute deviation after alignment), and (2) the ensemble TM-score (*TM-ens*). For the residue flexibility, both global (gl.) and per-target (pt.) mean/median correlations are reported; for the *TM-ens*, both mean/median are reported. Metrics are the higher the better ($\uparrow$).

| | Method | Apo/holo | | | Fold-switch | | |
|---|---|---|---|---|---|---|---|
| | | **ResFlex $r$ (gl.)** | **ResFlex $r$ (pt.)** | **TM-ens** | **ResFlex $r$ (gl.)** | **ResFlex $r$ (pt.)** | **TM-ens** |
| MSA-based | MSA-Subs. | 0.398 | 0.404 / 0.371 | 0.856 / 0.894 | 0.350 | 0.320 / 0.303 | 0.714 / 0.765 |
| | AlphaFlow | **0.455** | **0.527 / 0.527** | **0.864 / 0.893** | 0.385 | **0.384 / 0.376** | **0.730 / 0.788** |
| Seq-based | Eigenfold | 0.126 | 0.407 / 0.401 | 0.830 / 0.870 | 0.225 | 0.279 / 0.255 | 0.614 / 0.653 |
| | Str2Str (PF) | 0.174 | 0.326 / 0.307 | 0.731 / 0.728 | 0.161 | 0.246 / 0.233 | 0.615 / 0.644 |
| | Str2Str (SDE) | 0.148 | 0.349 / 0.340 | 0.659 / 0.681 | 0.111 | 0.224 / 0.220 | 0.521 / 0.545 |
| | ESMFlow | 0.416 | 0.496 / 0.522 | 0.856 / 0.893 | 0.269 | 0.345 / 0.329 | 0.700 / 0.755 |
| SLM | S-T5 | 0.097 | 0.144 / 0.166 | 0.726 / 0.787 | 0.313 | 0.135 / 0.099 | 0.437 / 0.392 |
| | S-GPT | 0.112 | 0.134 / 0.112 | 0.571 / 0.562 | 0.207 | 0.075 / 0.078 | 0.349 / 0.300 |
| | ESM3 (zero shot) | 0.312 | 0.473 / 0.466 | 0.839 / 0.876 | 0.388 | 0.323 / 0.320 | 0.627 / 0.717 |
| | ESMDiff (ID) | 0.424 | 0.502 / 0.517 | 0.851 / 0.883 | 0.391 | 0.328 / 0.346 | 0.660 / 0.720 |
| | ESMDiff (DDPM) | 0.420 | 0.489 / 0.515 | 0.838 / 0.877 | **0.402** | 0.341 / 0.288 | 0.626 / 0.685 |

## 5.2 CONFORMATION CHANGING PAIRS

We continue the study on the task of modeling and predicting conformational changes in structural proteins. The authors of Jing et al. (2023) have curated two benchmarking sets of pairing data including (1) 77 pairs of fold-switching proteins (Chakravarty & Porter, 2022); and (2) 90 apo/holo pairs with ligand-induced conformational change (Saldaño et al., 2022) to evaluate the modeling capacity of conformation diversity. Following the setting and evaluation metrics of Jing et al. (2023), we randomly sample a *few-shot* ensemble with five structures per target and evaluate them based on the correlation metrics of residue flexibility and the ensemble TM-score (Zhang & Skolnick, 2004). The evaluation results for both test sets are shown in Table 3. We find that MSA-based methods generally achieve better performance than other model families, which highlights the importance of using MSA for generating stable conformation changing protein targets.

Table 4: Mean absolute error (MAE) for the IDP test set measured in different characteristics for each method, where both mean / median are reported over all targets. All metrics are the lower the better (↓). The contact map of `PED00247` (PDB: 2MTF) is shown to the right as an example.

| | Method | Pairwise distance | Radius of gyration | Contact map |
|---|---|---|---|---|
| MSA-based | MSA-Subs. | 7.250 / 4.654 | 4.381 / 2.639 | **0.181 / 0.120** |
| | AlphaFlow | 7.129 / **3.533** | 4.880 / **1.464** | 0.228 / 0.161 |
| Seq-based | EigenFold | 11.419 / 6.404 | 8.663 / 4.880 | 1.309 / 0.560 |
| | idpGAN | 11.618 / 10.764 | 7.006 / 5.631 | 0.447 / 0.408 |
| | Str2Str (PF) | 9.838 / 7.301 | 6.369 / 4.286 | 0.264 / 0.203 |
| | Str2Str (SDE) | 8.793 / 5.891 | 5.444 / 3.082 | 0.227 / 0.165 |
| | ESMFlow | 7.692 / 4.429 | 5.110 / 1.636 | 0.257 / 0.179 |
| SLM | S-T5 | 9.009 / 5.737 | 5.448 / 2.596 | 0.536 / 0.467 |
| | S-GPT | 9.221 / 6.146 | 5.634 / 2.634 | 0.607 / 0.529 |
| | ESM3 (zero shot) | **6.606** / _4.301_ | 4.346 / 2.329 | _0.249_ / 0.174 |
| | ESMDiff (ID) | 6.886 / 4.689 | **4.333** / 2.160 | 0.295 / 0.239 |
| | ESMDiff (DDPM) | 7.010 / 4.665 | 4.438 / _1.974_ | 0.354 / 0.284 |

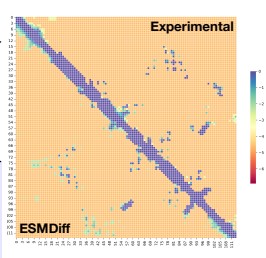

## 5.3 INTRINSICALLY DISORDERED PROTEINS

Different from structural proteins, intrinsically disordered proteins (IDPs) do not have a fixed or stable tertiary structure under normal conditions. IDPs possess inherent flexibility and usually exist as dynamic ensembles of conformations, allowing them to adapt to different binding partners or cellular environments. We have curated in total 114 entries from the protein ensemble database (PED) (Lazar et al., 2021) as benchmarking set. In specific, we only select the experimentally validated (eg. NMR spectroscopy) structure ensembles and excluding similar protein records with training set to avoid data leakage (see Appendix A.4). Due to the disordered structural characterization for IDPs, alignment-based metrics such as TM-score is not applicable. Because different targets have different sizes of ensemble (from ten to thousand), we follow the metrics used in Janson et al. (2023) to evaluate the mean absolute error (MAE), specifically the *pairwise distance*, *radius of gyration*, and *contact map* between the predicted ensemble and the ground-truth ensemble.

## 5.4 RUNTIME ANALYSIS

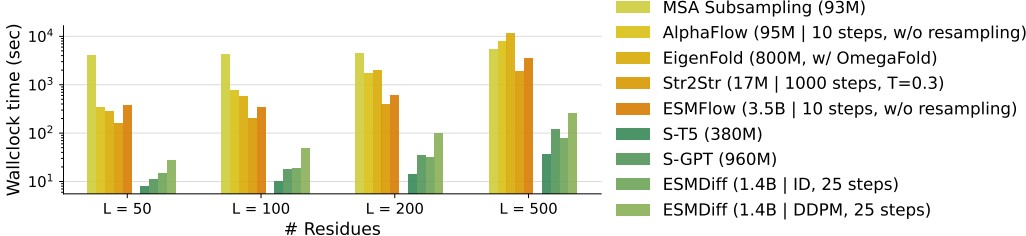

Figure 5: Runtime profiling for SLMs and other baseline methods. The number of parameters and necessary configurations for each model are also remarked for better reference.

To demonstrate the efficiency of SLM, we benchmark the runtime of each SLM and compare them with diffusion-based baselines. The measurement is based on the elapsed wall clock time for sampling an ensemble of $N = 100$ across different protein lengths (see Appendix A.2). As shown in Fig. 5, SLMs exhibit superior scalability with respect to protein size and are 20-100× faster than diffusion models like AlphaFlow, highlighting their potential for real-world applications.

## 5.5 Case Study: Structural Inpainting of Nanobody

The position-wise masking and unmasking nature inherently enables ESMDiff to perform the **inpainting** task effectively. Intuitively, the partially masked input (eg. nanobody framework) can be interpreted as an intermediate state $z_s, s > 0$ in the masked diffusion, where the unknown positions to be modeled are designated as [MASK] naturally in accordance with the formulation. Then partially reversing allows us to restrict the conformation sampling to the specific subregions of the protein, analogous to the concept of "inpainting" for images. Here we name the forward-backward inference as the **round-trip** masked diffusion inference (see Algorithm 5) for the inpainting task. Here, we illustrate the conformation inpainting capability through a case study that focuses on generating the complementarity-determining regions (CDRs) for a nanobody example. In specific, we explored the inpainting capabilities of ESMDiff using a nanobody derived from Llama (PDB entry: 1G9E).

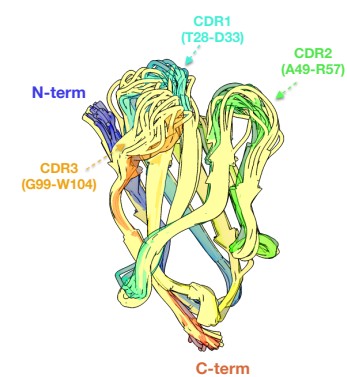

Figure 6: Visualization of a Llama VHH domain. The ESMDiff sampled conformations (in khaki) are superimposed on the NMR structures (Renisio et al., 2002) colored in transparent rainbow.

As shown in Fig. 6, the loop structures generated by the round-trip diffusion process of ESMDiff resemble the native conformations observed in NMR structures (both $N = 20$). The inference is conducted in a very efficient speed by only taking $\sim 5.3 \pm 0.3$ seconds on a single NVIDIA A100-SXM4-40GB GPU, highlighting the effectiveness of SLMs in performing structural inpainting and promising potentials in the real-world scenarios such as structural analysis or virtual screening.

## 6 Conclusion and Limitations

In this work, we propose a novel conformation generation framework of structure language models (SLMs). The overall inference is divided into two stages: conditional sampling of latent structure tokens and roto-translation invariant structure decoding. We develop and train a variety of conditional language models, especially the masked diffusion-based ESMDiff, which is a fine-tuned variant to enhance the capabilities of ESM3 adapting for conformation generation. Unlike existing methods, SLMs perform amortized distribution learning within an invariant latent space, leading to more efficient inference. By alleviating the need for geometric modeling, SLMs can fully exploit the scalability of modern language model architectures and take advantage of advanced hardware optimizations. Benchmarking results across various conformation generation tasks demonstrate the compelling performance and application potential of SLMs. In summary, the proposed method opens up an intriguing and novel research direction for related communities to explore.

*Limitations.* The current study presents several limitations worth exploring for future works. Firstly, one can design more advanced dVAE architecture to balance between structure disentanglement and reconstruction fidelity. In addition to the discrete latent space, continuous latent space can also be considered, for example using the latent diffusion models (Rombach et al., 2022). Secondly, it is worthwhile exploring alternative SLM instances specially tailored for the sequence-to-structure translation in consideration of proper inductive biases. Additionally, modeling protein side chains and other relevant modality including ligand and RNA by building an atomic structural auto-encoder is also a very appealing research direction. Lastly, the outstanding performance of MSA-based methods also indicates the potential to build MSA-conditioned structure language models.

## REPRODUCIBILITY STATEMENT

For reproducibility, we provide detailed implementation specifics, including the baseline inference pipeline and training/fine-tuning procedures in Appendix A. The sampling pipelines for ESM3 and ESMDiff are thoroughly described in Appendix B, while Appendix D outlines the evaluation methods and their references. The theoretical results mentioned in the main text are derived or proven in Appendix G.2. The source training and inference code for structure language models in this study are made publicly available at `https://github.com/lujiarui/esmdiff`, as we believe fine-tuning foundation protein language models can have broader applicability in various downstream tasks including the protein conformation generation.

## ACKNOWLEDGMENTS

We thank Can Chen and Chenqing Hua for helpful feedback as well as anonymous reviewers for their constructive suggestion and comments. This project is supported by the Natural Sciences and Engineering Research Council (NSERC) Discovery Grant, the Canada CIFAR AI Chair Program, collaboration grants between Microsoft Research and Mila, Tencent AI Lab Rhino-Bird Gift Fund and a NRC Collaborative R&D Project (AI4D-132). This project was also partially funded by IVADO Fundamental Research Project grant PRF-2019-3583139727. YB acknowledges funding from NRC AI4D, CIFAR and CIFAR AI Chair. The computation resource of this project is supported by Mila, the Digital Research Alliance of Canada, and NRC.

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

# A    IMPLEMENTATION DETAILS

## A.1    BASELINE EVALUATIONS

For baselines, we compare our structure language models against: EigenFold (Jing et al., 2023), Str2Str (Lu et al., 2024b), MSA subsampling (Del Alamo et al., 2022), AlphaFlow & ESM-Flow (Jing et al., 2024a), and idpGAN (Janson et al., 2023). For MSA subsampling, we leverage the official repository of Del Alamo et al. (2022) under AlphaFold v2.3.2 with an MSA depth of 32, i.e. 32 maximum extra MSAs and 16 maximum MSA clusters (half the depth value (Del Alamo et al., 2022)). For AlphaFlow and ESMFlow, we use the base PDB model (no distillation) with no resampling. The input MSAs for test proteins are queried via the ColabFold server (Mirdita et al., 2022). All other parameters are set to the recommended values according to the original implementation by the authors (Jing et al., 2024a). For EigenFold, we follow the pipeline in Jing et al. (2023) by using OmegaFold (Wu et al., 2022) to make embeddings for the test set and sampling via the pre-trained diffusion model with default parameters, where $\alpha = 1$ and $\beta = 3$. For Str2Str, we use their public codebase to with initial structure guess generated by the ESMFold (Lin et al., 2023) using the script provided in the repository. We use a $T = 1000$ diffusion sampling steps for both probability flow and SDE (noise scale is set to 1.0 for SDE runs) and other parameters as default. For idpGAN, we run the inference pipeline following the Jupyter notebook in their open-source repository with the default inference settings for custom proteins.

## A.2    RUNTIME PROFILING

To profile the inference time for each method, we set a ladder of tasks of different protein lengths including 50, 100, 200, and 500 residues, with an ensemble size of $N = 100$ for each benchmark task. The profiling is carried out on a single NVIDIA A100 SXM4 GPU with 40GB memory, we measure the wall clock time elapsed for generating the target ensembles for each model. Note that for the MSA-based methods, the time-consuming MSA search is conducted only once and is amortized in the overall computational cost for multiple batches. For autoregressive SLMs namely S-T5 and S-GPT, the past computed key-value pairs for each layer are cached for fast inference.

## A.3    TRAINING DETAILS

**S-T5 training.**    For S-T5, we build upon the implementation from the Transformers package (Wolf et al., 2020). Table S1 shows the hyperparameter settings for training. The total number of trainable parameters is 384M. The model is trained without learning rate scheduler for up to 30 epochs. The final tested model and the reported epoch number in Table S1 come from the best checkpoint, selected according to the NLL of structure tokens on a hold-out validation set. To leverage the ESM3 sequence embeddings, we bypass the encoder's input token embedding layer and feed the precomputed embeddings directly into the transformer encoder.

**S-GPT training.**    For S-GPT, we also use the implementation from the Transformers package. The total number of trainable parameters is 961M. We insert a special [sep] token in between the sequence and structure tokens when they are fed into the model, with its embedding being randomly initialized and learnable. In practice, we find it useful to add a hyperparameter $\lambda$ to balance between the NLL of sequence and structure prediction, i.e., $\mathcal{L}(\theta) = -\mathbb{E}_{(\boldsymbol{c},\boldsymbol{x})\sim\mathcal{D}}\mathbb{E}_{\boldsymbol{z}\sim q(\boldsymbol{z}|\boldsymbol{x})}\left[\lambda\sum_{l=1}^{L}\log p_\theta(\boldsymbol{y}_l|\boldsymbol{y}_{<l}) + \sum_{l=L}^{2L}\log p_\theta(\boldsymbol{y}_l|\boldsymbol{y}_{<l})\right]$, where the first half of $\boldsymbol{y}$ consists of the sequence tokens and the second half consists of the structure tokens, as defined in Sec. 3.2. Table S2 shows the hyperparameter settings for training. The model is trained and up to 30 epochs but the best checkpoint is evaluated and reported.

**Masked Diffusion fine-tuning of ESM3.**    The masked diffusion is scheduled with a log linear noise schedule (Austin et al., 2021) for clipped time domain $t \in [0, 1-\epsilon]$ with infinite small time $\epsilon = 0.001$, such that $\sigma(t) \triangleq -\log(1-t)$ while the coefficient is parameterized as $\alpha(t) = \exp[-\sigma(t)] \in (0, 1]$. We leverage this schedule for both training and inference. When fine-tuning, we replace the pre-trained LM head in the structure track of the ESM3 with a randomly initialized head for the augmented vocabulary $\bar{V} = V \cup [\text{MASK}]$ by setting output dimension to be 4101. We use a learned embedding module to encode timestep $t$ via the sinusoidal time embeddings followed

Table S1: Hyperparameters of S-T5

| Parameter | Value | Description |
|---|---|---|
| optimizer | AdamW | Optimizer type |
| lr | $1 \times 10^{-5}$ | Learning rate |
| batch_size | 8 | Batch size |
| epochs | 28 | Number of training epochs |
| betas | (0.9, 0.999) | Coefficients for computing running averages of gradient |
| weight_decay | $1 \times 10^{-2}$ | Weight decay coefficient |
| num_layers | 12 | Number of transformer layers in the encoder and decoder |
| num_heads | 16 | Number of attention heads |
| d_ff | 2048 | Feedforward network dimension |
| d_model | 1280 | Model hidden dimensionality |
| dropout_rate | 0.1 | Dropout rate |
| feed_forward_proj | gelu | Activation function |

Table S2: Hyperparameters of S-GPT

| Parameter | Value | Description |
|---|---|---|
| optimizer | AdamW | Optimizer type |
| lr | $1 \times 10^{-4}$ | Learning rate |
| batch_size | 12 | Batch size |
| epochs | 11 | Number of training epochs |
| betas | (0.9, 0.999) | Coefficients for computing running averages of gradient |
| weight_decay | $1 \times 10^{-2}$ | Weight decay coefficient |
| num_layers | 48 | Number of transformer layers |
| num_heads | 16 | Number of attention heads |
| n_inner | 5120 | Feedforward network dimension |
| n_embd | 1280 | Model hidden dimensionality |
| dropout_rate | 0.1 | Dropout rate |
| activation_function | gelu | Activation function |
| seq_pred_weight | 0.01 | Weight of loss for sequence prediction |

a simple 3-layer MLP coupled with SiLU activation function. We use the AdamW optimizer with betas=(0.9, 0.999) and weight decay=0.01. The training is scheduled with the constant scheduler with warm up steps of 2,500.

---

**Algorithm 2** Masked diffusion fine-tuning of ESM3
___

1: **Require:** Pre-trained masked language models $p_\theta(\cdot|c, z)$, training data $\mathcal{D} = \{c, z\}$, masking schedule $\alpha(t) > 0$, learning rate $\eta$.
2: **for** $i = 1, 2, \ldots, |\mathcal{D}|$ **do**
3:      Get sample $(c, z) \leftarrow \mathcal{D}[i]$
4:      Sample time indices $t \in U[0, 1]$
5:      Sample masked probability $p(t)$ based on the masking schedule $\alpha(t)$
6:      Create masked input $z_t$ by masking $z$ with probability $p(t)$
7:      Compute the negative ELBO loss $\mathcal{L}(\theta; z_t, t, c)$ based on Eq. 3
8:      Update model parameters: $\theta \leftarrow \theta - \eta \nabla_\theta \mathcal{L}(\theta)$
9: **return** $p_\theta$

---

## A.4 CURATION OF IDP TEST SET

We curated the test set for intrinsically disordered proteins (IDPs) by downloading data from the Protein Ensemble Database (PED) (Lazar et al., 2021) on August 10, 2024, using the official API, which provided a total of 481 raw entries. To prepare the evaluation set, we filtered the data to include proteins with sequence lengths between 20 and 500 and ensemble sizes of $N \geq 10$. To ensure balance in the dataset, we performed clustering using MMseqs 2 (Steinegger & Söding, 2017) with the flags -s 7.5 --min-seq-id 9.0. We further excluded the records with more

than 90% sequence identity by conducting another search against the PDB training data to avoid data leakage. Finally, we select the experimentally validated (eg., NMR) ensembles by filtering with respect to the measurement method, which excludes computationally generated ensemble by models such as idpGAN (Janson et al., 2023).

## A.5 TRAINING DATA OF BASELINE MODELS

We list the training data and the corresponding cutoff for all the mentioned models in our study for reference, as shown in Table S3.

Table S3: Training data and base model of different methods in this study.

| Method | Base/Embedding model | Data source | Cutoff date |
|---|---|---|---|
| MSA subs. | AlphaFold2 (93M) | PDB | May 1, 2018 |
| AlphaFlow | AlphaFold2 (93M) | PDB | May 1, 2018 |
| ESMFlow | ESMFold (3B) | PDB | May 1, 2020 |
| Eigenfold | OmegaFold (900M) | PDB | Apr. 30, 2020 |
| Str2Str | / | PDB | Jun, 9, 2023 |
| idpGAN | / | DisProt50 + MD | Jun., 2021 |
| ESMDiff | ESM3-open (1.4B) | PDB | May 1, 2020 |

## B INFERENCE OF BIDIRECTIONAL SLMS

### B.1 ZERO-SHOT CONFORMATION GENERATION OF ESM3

The sampling method for the BERT model was first introduced by Wang & Cho (2019), which treats the bidirectional encoder as a Markov random field graphical model and employs ID sampling to generate new sequences. In Hayes et al. (2024), the author propose to use iterative decoding to enable simultaneous decoding of multiple positions rather than processing them one at a time. Following the authors (Hayes et al., 2024), we refer to this latter method as the iterative decoding (ID) and take advantages of this sampling strategy for zero-shot conformation generation.

In specific, we leveraged the open-source 1.4B ESM3 protein language model (Hayes et al., 2024) for zero-shot conformation sampling. ESM3 is pre-trained on multi-modal data related to protein and contains several tracks of language model heads for prediction as well as input embeddings, including sequence (amino acid types), structure, secondary structure, SASA and functions. In our main experiment, only the sequence and structure tracks are turned on and the input tokens for other tracks are set to the default value (i.e. [MASK] in each vocabulary). For sampling configurations, we use a sampling schedule of 25 steps and set the sampling temperature to be 1.0 without specially pointing out. During sampling, we also adopted the nucleus (top-p) sampling strategy (Holtzman et al., 2019) to improve the quality of samples with a probability threshold of 0.95. Throughout the experimental Section 5, we use the entropy-ranked ID sampling to obtain the results for ESM3 (described below).

### B.2 POSITION-RANKED ITERATIVE DECODING

Unlike one-shot decoding, which maps all masked structure tokens [MASK] to their predictions in a single feed-forward pass, the iterative decoding can adopt various strategies to iteratively select where to unmask at each sampling step from the masked structure tokens of the target protein. These strategies are governed by different ranking functions $f(\cdot)$ applied over the masked positions, as detailed below. The overall inference pipeline (with ranking) is outlined in Algorithm 3. We investigate and compare these ranking strategies in Table S4, using ESM3 as the base model.

**Entropy ranking.** The entropy for categorical distribution quantifies the predicted uncertainty for the current evaluation of probability. Formally, the ranking score of the masked position indexed by $i$ $(1 < i < N)$ is the position-specific entropy:

$$f(i) = -\sum_{z' \in V} p_{[i]}(z_{[i]} = z') \log p_{[i]}(z_{[i]} = z'), \tag{4}$$

---

**Algorithm 3** Iterative Decoding with Positional Ranking

---

1: **Require:** amino acid types (condition) $c$, masked language models $p_\theta(z|c, z)$, ranking function $f(i; c, z)$, sampling temperature $T > 0$, the number of decoding steps $K$, mask token [MASK] .
2: $L = \text{len}(c)$                      ▷ Target protein length
3: Initialize $M \leftarrow \{0, 1, \ldots, L - 1\}, U \leftarrow \varnothing$
4: Initialize $z_{[i]} \leftarrow$ [MASK] $, \forall i \in M$
5: Get schedule $\{n_k\}$ $(k = 1, \ldots, K)$ with $\sum_k n_k = L$ ▷ per-step number of positions to unmask
6: **for** $k = 1$ to $K$ **do**                               ▷ iterations
7:      Rank $i \in M$ using $f(i; c, z)$
8:      Select top $n_k$ positions $M_k \leftarrow \text{Ranked}(M; key = f)[: n_k]$        ▷ rank and select top-$n_k$
9:      Sample latent components $z'_{[i]} \sim p_\theta(z_{[i]}|c, z), i \in M_k$ with temperature $T$
10:      Assign $z_{[i]} \leftarrow z'_{[i]}, \forall i \in M_k$
11:      Update the unmasked set: $U \leftarrow U \cup M_k$
12:      Update the masked set: $M \leftarrow M \setminus M_k$
13: **return** $z$

---

where $p_{[i]} = \text{SoftMax}(\text{logits}_{[i]})$ is the predicted probability mass function of position $i$ over the pre-defined vocabulary of valid structure tokens $V$. The entropy of each candidate position is ranked ascending, i.e. **positions with smaller entropy are firstly decoded**. We also adopt the adaptive entropy calculation, which means the logits prediction in the current decoding step is based on the unmasked structure tokens from the last step and calculated on-the-fly.

**Maxlogit ranking.** Beside the uncertainty-based ranking, we similarly consider the maximum of logits (*maxlogit*) from prediction as the ranking score. This indicates we choose **the positions with the top (highest) logit score as candidates**. The maxlogit reflects the maximal confidence of the model among its choice as well as the probability of the selected token during greedy decoding. Correspondingly, the ranking score of maxlogit for index $i$ is defined as:

$$f(i) = \max_{z' \in V} p_{[i]}(z_{[i]} = z').$$
(5)

**Secondary structure ranking.** The ranking scores discussed above can be applied to arbitrary data for masked decoding. Here, we introduce a novel decoding schedule designed for proteins, ordered by predicted secondary structure:

$$f(i) = -g(s^*_{[i]}), \text{where } s^*_{[i]} = \arg\max_{s^* \in \mathcal{S}} p_{ss}(s_*|c, z, i),$$
(6)

where $p_{ss}$ is a predicted categorical distribution over all eight secondary structures (SS8) defined by the Dictionary of Protein Secondary Structure (DSSP), including *helix* of 3-turn, 4-turn, 5-turn, and hydrogen bonded turn; extended strand of $\beta$-*sheet* and isolated $\beta$-*bridge*; *bend* and *coil*, and $g(\cdot) \in \{0, \ldots, 7\}$ maps the SS8 type to the finite ordering score. Given the predicted distribution over SS8, we use greedy selection to assign the SS8 label (with the max logit) for all masked positions. The positions are further ranked from the most *structural* to the most *disordered*, or the same order as above. The intuition behind is that we want to predict the structural region in the first place, followed by modeling the disordered region (loop). In practice, we adopt the secondary structure prediction heads of pre-trained ESM3 (Hayes et al., 2024) for the prediction of SS8 types for each masked candidate positions.

## B.3    INFERENCE OF CONDITIONAL MASKED DIFFUSION MODEL

After the fine-tuning stage, ESMDiff is able to perform conditional generation from sequence to structure either using (1) the iterative decoding described in Algorithm 3 or (2) the vanilla DDPM ancestral sampling (Ho et al., 2020). The former sampling pipeline can readily accommodate the ESMDiff backbone (fine-tuned ESM3 Transformer module) without any changes. For experiments, we fine-tune the ESM3 using the objective defined in Eq. 3 w/o time conditioning. The DDPM sampling of masked diffusion model is simple and straightforward by progressively unmasking all the [MASK] in a fixed number of time steps $T = 25$. This inference pipeline is shown in Algorithm 4.

Table S4: The performance of different ranking functions on the three benchmarking tasks described in Appendix B.2.

| Method | JS-PwD ($\downarrow$) | JS-TIC ($\downarrow$) | JS-RG ($\downarrow$) | Validity ($\uparrow$) | TM-ens ($\uparrow$) | RMSD-ens ($\downarrow$) |
|---|---|---|---|---|---|---|
| Uniform (w/o ranking) | 0.414 | 0.428 | 0.602 | 0.840 | 0.845 | 1.460 |
| Entropy | **0.411** | **0.402** | 0.582 | 0.740 | 0.844 | 1.478 |
| Maxlogit | 0.411 | 0.425 | **0.576** | 0.890 | **0.850** | **1.436** |
| Secondary structure | 0.411 | 0.423 | 0.628 | **0.930** | 0.849 | 1.441 |

| Method | Apo/holo | | | Fold-switch | | |
|---|---|---|---|---|---|---|
| | ResFlex $r$ (gl.) | ResFlex $r$ (pt.) | TM-ens | ResFlex $r$ (gl.) | ResFlex $r$ (pt.) | TM-ens |
| Uniform (w/o ranking) | 0.223 | 0.400 / 0.384 | 0.796 / 0.848 | 0.328 | 0.276 / 0.291 | 0.531 / 0.522 |
| Entropy | 0.318 | 0.447 / 0.470 | 0.840 / 0.876 | **0.407** | **0.343 / 0.366** | **0.629** / 0.692 |
| Maxlogit | **0.386** | **0.480 / 0.481** | **0.843 / 0.876** | 0.391 | 0.333 / 0.309 | **0.629 / 0.715** |
| Secondary structure | 0.237 | 0.444 / 0.444 | 0.826 / 0.868 | 0.377 | 0.309 / 0.352 | 0.589 / 0.665 |

| Method | Pairwise distance | Radius of gyration | Contact map |
|---|---|---|---|
| Uniform (w/o ranking) | 6.661 / 4.810 | 4.017 / 1.950 | 0.328 / 0.264 |
| Entropy | 6.706 / **4.726** | 4.273 / 2.172 | **0.249 / 0.170** |
| Maxlogit | 6.743 / 5.037 | 4.311 / 2.303 | 0.252 / 0.191 |
| Secondary structure | **6.365** / 4.767 | **3.833 / 2.011** | 0.298 / 0.228 |

The DDPM sampling is conducted for fine-tuned model w/ time conditioning. Throughout our study, we neither control the temperature (equivalently set to 1.0) nor use the ranking function introduced in Appendix B.2 during each reverse step in DDPM sampling, which is worth exploring in the future works.

---

**Algorithm 4** DDPM Ancestral Sampling for Conditional Masked Diffusion

---

1: **Require:** amino acid types (condition) $c$, bidirectional model (w/o softmax head) $f_\theta(z_s|z_t, c)$, noise schedule $\{\alpha_t\}$, initial latent $z_T \sim q(z_T)$, the number of denoising steps $T$, mask token [MASK]
2: $L = \text{len}(c)$          $\triangleright$ Target protein length
3: Initialize $z_T \leftarrow [0, \dots, 0] \in |\bar{V}|^L$      $\triangleright$ Initialize with noisy latent vector from prior
4: Set mask tokens: $z_{T,[i]} \leftarrow [\text{MASK}], \forall i \in \{1, 2, \dots, L\}$
5: **for** $t = T$ to $1$ **do**        $\triangleright$ Ancestral denoising process
6:      $P^*_{L \times |\bar{V}|} \leftarrow \text{Softmax}(f_\theta(z_{t-1}|z_t, c))$      $\triangleright$ Compute log probability matrix from the logits
7:      For each $i$, $z^*_{[i]} \sim \text{Cat}(\cdot; P^*[i,:])$    $\triangleright$ Sample candidate tokens from categorical distribution
8:      $z^*_{t-1} \sim q(z_{t-1}|z_t, z^*)$        $\triangleright$ Sample from the posterior according to Equation 2
9:      **for** $i = 1$ to $L$ **do**
10:          **if** $z_{t,[i]} = [\text{MASK}]$ **then**
11:             $z_{t-1,[i]} \leftarrow z^*_{t-1,[i]}$        $\triangleright$ Update using sampled $z^*_{t-1}$
12:          **else**
13:             $z_{t-1,[i]} \leftarrow z_{t,[i]}$        $\triangleright$ Preserve the unmasked token from $z_t$
14: **return** $z_0$        $\triangleright$ Return the fully unmasked latent vector

---

### B.4 COMPARISON BETWEEN ESM AND ESMDIFF

Different from the MLM pre-training of ESM, in ESMDiff, the amino acid types are fully conditioned while the structure tokens are trained using masked diffusion with loglinear schedule. Except for the time-conditioning and loss reweighting, both MLM and masked diffusion similarly use random masking as training objective. However, the ratio of masking should depend on different downstream tasks to "mimic the types of inputs" during inference, as indicated in Hayes et al. (2024). Thus, in the task of SLMs, we reason that it benefits conformation generation more when using less-to-none sequence masking and properly scheduled structure masking. For other fine-tuning application of ESM (eg., sequence to function tokens), we recommend practitioners to make the fine-tuning objective suitable for specific input/output inference-time usage.

## C   DETAILS OF CONFORMATION INPAINTING

The inherently position-wise masking and unmasking nature of masked diffusion enables ESMDiff to perform inpainting tasks effectively. Intuitively, the partially masked input can be interpreted as an intermediate state $z_s(s > 0)$ within the masked diffusion process, where the unknown positions that need to be modeled are designated as [MASK] in accordance with the masked diffusion formulation. This allows us to restrict the sampling of conformational changes to specific sub-regions of protein structures, analogous to the concept of "inpainting" in image processing.

Specifically, the conformation inpainting can be formulated as a *round-trip* masked diffusion process described below:

**Forward.**   Given an input sequence of $N$ tokens $z_0 \in |V|^N$ as the initial state, we first simulate the forward diffusion process to an intermediate state $z_s$. Instead of using the stochastic forward kernel $q(z_s|z_0)$ defined in Eq. 9, $z_s$ is obtained in a deterministic way, i.e., we set $z_{s,[i]} \leftarrow$ [MASK] if $\forall i \in \Delta_M$ and $z_{s,[i]} \leftarrow z_{0,[i]}$ otherwise, where $\Delta_M$ is the residue indices for which we want to sample. Note that the ideal time $s$ is induced from the noise schedule $\alpha(t)$ by matching the expectation of the number of the masking positions under the forward marginal, or formally, $s^* = \alpha^{-1}(N - |\Delta_M|)$. Intuitively, we shall see on average $n = |\Delta_M|$ [MASK] tokens in $z_s^*$ if we simulate the stochastic kernel $z_{s^*} \sim q(\cdot|z_0)$. In practice, however, $s^* > 0$ can be chosen as a tunable hyperparameter.

**Backward.**   We then evolve the reverse process of masked diffusion starting using the learned posterior $p_\theta(z_s|z_t, c)$ from $z_s$ according to Eq. 2. Thanks to the "absorbing" nature of the masked diffusion process, the copied variables $\{z_{s,[i]}, \forall i \notin \Delta_M\}$ are all preserved during the backward steps until we return to $\hat{z}_0$ (we add hat to distinguish it from input $z_0$). This means that sampling is conducted exclusively for the target indices in $\Delta_M$.

Since the sampling transitions from the fully unmasked state $z_0$ to $z_s$ ($s > 0$) and then back to $\hat{z}_0$, we refer to this as the "*round-trip*" diffusion process. This approach is reminiscent of the forward-backward (FB) dynamics introduced by Lu et al. (2024b), although the latter is defined for local exploration within the $\mathrm{SE}(3)^N$ geometric space and is not specifically intended for inpainting. We describe this process in Algorithm 5.

---

**Algorithm 5** Round-Trip Diffusion for Conformation Inpainting

---

1: **Input:** Initial structure $x$, amino acid tokens $c$, the indices of the sampling sub-region $\Delta_M$, noise schedule $\alpha(t)$, denoising network $u_\theta$, structure encoder $q_\psi$, and structure decoder $p_\phi$
2: **Output:** The inpainted conformation $\hat{x}$
3: Encode $z_0 \sim p_\psi(z_0|x)$
4: Length $L \leftarrow \mathrm{len}(z_0)$
5: // Deterministic Forward Diffusion
6: Infer the ideal intermediate time step $s^* = \alpha^{-1}(N - |\Delta_M|)$
7: Set $z_s^* \leftarrow z_0$
8: **for** each $i \in \Delta_M$ **do**
9:      $z_{s^*,[i]} \leftarrow$ [MASK]                                        ▷ Mask specified indices
10: // Generating Reverse Diffusion
11: $\hat{z}_0 \leftarrow \mathrm{Reverse}(z_s^*, c, u_\theta)$                                        ▷ Either in Algo. 3 or 4
12: Decode $\hat{x} \sim p_\phi(x|c, \hat{z}_0)$
13: **return** $\hat{x}$

---

In Fig. 6, we explored the inpainting capabilities of ESMDiff using a nanobody derived from Llama (PDB entry: 1G9E). To sample the conformations of the CDR loops, we began with a PDB structure, fixed the nanobody framework, and applied the inpainting pipeline exclusively to the three CDR loops, which is numbered by the canonical IMGT numbering scheme (Lefranc et al., 2003). Then we simulate the round-trip diffusion process using ESMDiff (ID, T=1.0) with 25 steps for the CDR regions. The sampling is performed on a single NVIDIA A100-SXM4-40GB GPU. Both NMR and the generated structure ensembles contain $N = 20$ conformations.

## D  EVALUATION METRICS

**Jensen-Shannon divergence (JS).**    The quality of a set of conformations is assessed by comparing the distributional similarity between the reference ensemble and the generated ensemble, akin to how the Fréchet inception distance (FID) (Heusel et al., 2017) is used for evaluating synthetic images. We use the Jensen-Shannon (JS) divergence because of its symmetry, which penalizes the model's distribution for both lack of ground truth coverage and biased coverage. To ensure compatibility with baseline models, only $C^\alpha$-atoms are considered in calculating divergence metrics. We use three key roto-translation invariant features to accurately capture ensemble characteristics, following Lu et al. (2024b):

- Pairwise distance (PwD): pairwise distances are calculated between atoms, skipping three atoms between pairs. To transform these continuous values into a distribution, we construct histograms with $N_{\text{bin}} = 50$ bins to represent the pairwise distribution, and JS divergence is calculated over these. A pseudo-count of $\epsilon = 10^{-6}$ is added to zero frequencies for slight smoothing.

- The two slowest components of time-lagged independent component analysis (TIC) (Pérez-Hernández et al., 2013): Pairwise distances for each protein conformation are computed and flattened. TICA projections are then fitted using the reference full MD trajectories, applying the Deeptime library(Hoffmann et al., 2021). The first two components are selected after TICA dimension reduction for each ensemble. Histograms are constructed for both components similarly to the pairwise distances.

- Radius of gyration: This measures the root mean square distance of atoms from the center of mass. The same histogram approach is used as described for the other features.

**Validity.**    following Lu et al. (2024b), the validity is defined as the ratio of clash-free conformations in the generated ensemble. It is computed by dividing the number of conformations without steric clashes by the total size of ensemble. A steric clash occurs when two atoms are too close to each other. For a given ensemble of conformations, it is expressed as: $\text{Validity}(\{\boldsymbol{x}^{(i)}\}_{i=1}^N) = 1.0 - \frac{1}{N}\sum_{i=1}^N \mathbf{1}\{\exists\, j, k, \text{such that } |\boldsymbol{x}_{C^\alpha,[j]}^{(i)} - \boldsymbol{x}_{C^\alpha,[k]}^{(i)}| < \delta\}$, where $\boldsymbol{x}_{C^\alpha,[j]}^{(i)} \in \mathbb{R}^3$ represents the $C^\alpha$-atom coordinate of residue $j$ in the conformation sample $\boldsymbol{x}^{(i)}$, and $\delta = 3.0\text{Å}$.

**TM-ens (RMSD-ens).**    First introduced in Jing et al. (2023), the ensemble TM-score (TM-ens) quantifies how well the generated ensemble matches the observed conformational states in the reference data. Specifically, it is defined as the maximum TM-score between each generated sample and the reference state, averaged over all reference states. This provides a measure of structural similarity, with higher scores indicating better coverage between the generated and true conformations given a limited sampling budget. Similarly, RMSD-ens calculates the average minimum RMSD value between the generated and reference conformations, providing another metric to evaluate the structural fidelity of the ensemble. They are respectively defined as follows, with $\boldsymbol{x}$ represents the generated ensemble and $\boldsymbol{y}$ denotes the reference ensemble:

$$\text{TM}_{\text{ens}}(\{\boldsymbol{x}^{(i)}\}, \{\mathbf{y}^{(j)}\}) = \frac{1}{|\{\mathbf{y}^{(j)}\}|} \sum_{j=1}^{|\{\mathbf{y}^{(j)}\}|} \max_i \text{TM}(\boldsymbol{x}^{(i)}, \mathbf{y}^{(j)}), \text{ and}$$

$$\text{RMSD}_{\text{ens}}(\{\boldsymbol{x}^{(i)}\}, \{\mathbf{y}^{(j)}\}) = \frac{1}{|\{\mathbf{y}^{(j)}\}|} \sum_{j=1}^{|\{\mathbf{y}^{(j)}\}|} \min_i \text{RMSD}(\boldsymbol{x}^{(i)}, \mathbf{y}^{(j)}).$$

The TM-score and RMSD are both calculated using the compiled binary executive developed by Zhang & Skolnick (2004).

**Residue flexibility.**    Following Jing et al. (2023), we evaluate the Pearson correlation for global and per-target residue flexibility (ResFlex), which is defined as the absolute deviation of the center

$C^\alpha$-atom coordinates after structural alignment, or formally for index $i$:

$$\text{ResFlex}(\{\boldsymbol{x}_k\}_{k=1}^N)[i] = \frac{1}{N}\sum_{k=1}^N |\text{Align}(\boldsymbol{x}^{(k)})_{C^\alpha,[i]} - \frac{1}{N}\sum_{k=1}^N \text{Align}(\boldsymbol{x}^{(k)})_{C^\alpha,[i]}|,$$

where $\text{Align}(\cdot)$ means the Kabsch alignment as in RMSD among all $N$ conformations in the ensemble. Afterward, the correlation is calculated between the generated ensemble and the reference ensemble.

**Mean absolute error (MAE).** Since the experimental ensemble contains a varying number of samples for different targets in the PED database (Lazar et al., 2021), we balance by using the mean absolute error (MAE) to evaluate the performance on intrinsically disordered proteins (IDPs). For each feature, we calculate the ensemble average for both the reference and generated samples, then compute the MAE between them and average this value over the channels. The mean and median MAE are reported across all 114 targets.

**Contact map.** We calculate and visualize the contact map as the probability distribution over a 2D grid of pairwise distances. First, the pairwise distance between residues is calculated, and a distance threshold of $8.0\text{Å}$ is applied to determine if two residues are in contact (Janson et al., 2023). Finally, the frequency of contacts across all conformations in the ensemble is normalized to create a probability distribution over the entire distance map. Formally, the contact probability between residues $i$ and $j$ is given by:

$$P_{\text{contact}}(i,j) = \frac{1}{N}\sum_{k=1}^N \mathbf{1}\{d^{(k)}(i,j) \leq 8.0\text{Å}\},$$

where $d^{(k)}(i,j) \in \mathbb{R}$ is the pairwise distance between residues $i$ and $j$ in conformation $k$, and $N$ is the number of conformations in the ensemble. This normalized contact map provides straightforward insights into residue interactions within the generated ensemble.

# E  ABLATION STUDIES AND ADDITIONAL EXPERIMENTS

## E.1  FINE-TUNING CONFIGURATION

In Table S5, we list different fine-tuning configurations in our experiments. Note that using the sequence track dropout, masking, or prediction similar to the pre-training stage of ESM3 do have a positive improvement on the BPTI task, we discover the model accuracy will decrease for the apo/holo or fold-switch evaluation. In specific, the track dropout rate is set to be 25%, the masking rate is 10% while the sequence prediction loss has the equal weight of structure prediction.

## E.2  CHANGING OF TEMPERATURES

In this section, we examine the effect of altering the temperature parameter $T$ during the sampling process. Temperature scaling influences the diversity and accuracy of the sampled conformations, with higher temperatures encouraging more exploratory behavior, while lower temperatures lead to more deterministic outputs. We experimented with different values of $T$, ranging from 0.25 to 5.0, to observe how this affects the balance between diversity and accuracy in our conformation generation for different SLMs (for ESMDiff, only the ID sampling is applied). To make the result straightforward, we evaluate on the BPTI samples (N=100) with the validity metric as well as the TM-diversity. The TM-diversity is defined as the average pairwise inverse TM-score (i.e., $1 - \text{TM}(x,y)$) among conformations in the sampled ensemble. As shown in Fig. S1, we observe that higher temperature generally leads to decreased validity and increased diversity. Note that the pre-trained ESM3's (w/o fine-tuning) performance degenerated significantly at the low temperature. After fine-tuning, its validity improves significantly without compromising the diversity. As the temperature increases, ESM3 and ESMDiff maintain a high validity score. However, there is minimal improvement in diversity, suggesting that these models might not be sensitive to high sampling temperatures. S-T5 and S-GPT also achieve strong validity when $T \leq 1.0$. In contrast to ESM models, these two models are quite sensitive to the temperature. As can be seen, their validity scores quickly deteriorate as

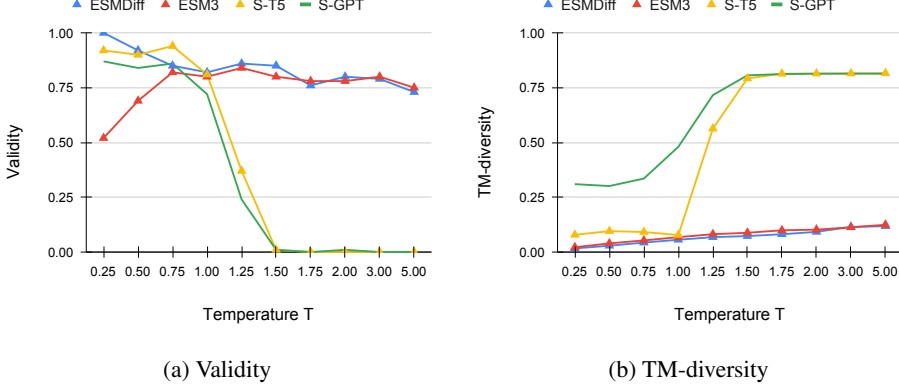

(a) Validity  (b) TM-diversity

Figure S1: Moving (a) Validity and (b) TM-diversity along with temperature for SLMs.

the temperature moves beyond 1.5. Notably, at $T \leq 1.0$, S-GPT not only maintains high validity but also achieves significantly greater diversity, showcasing its superior ability to balance between diversity and accuracy.

### E.3 CHANGING OF SAMPLING STEPS

This section focuses on the impact of varying the number of sampling steps in the ID or DDPM sampling. As the number of steps increases, the model has more opportunities to refine the sampled conformations, potentially improving accuracy. However, increasing the total number of sampling steps $K$ also incurs a higher computational cost (growing linearly $O(K)$). We explored different step counts, ranging from 5 to 100, and evaluated their performance similarly. As shown in Fig. S2, only ID sampling significantly benefits from an increased number of sampling steps. For the other two methods, accuracy improvement plateaus after 10 steps. The diversity remains largely unaffected by the number of steps for all methods. Overall, DDPM sampling consistently yields the highest validity across nearly all step counts, with the exception of at 100 steps.

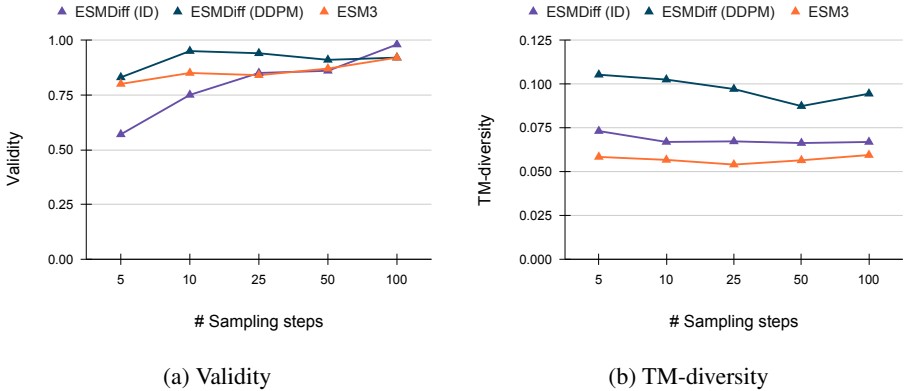

(a) Validity  (b) TM-diversity

Figure S2: Moving (a) Validity and (b) TM-diversity along with the number of sampling steps for BERT-like SLMs.

### E.4 EVALUATING ON THE ATLAS MD ENSEMBLES

Unlike Boltzmann generators (Noé et al., 2019; Arts et al., 2023), recent studies have focused on developing generative models that learn from PDB data plus multiple molecular dynamics (MD) trajectories (Zheng et al., 2023b; Jing et al., 2024a;b; Lu et al., 2024a). We further evaluate performance on the ATLAS MD ensemble dataset (Vander Meersche et al., 2024), which comprises 100ns all-atom molecular dynamics simulations across 1,390 monomeric protein targets. In Jing

Table S5: Ablation results for ESMDiff fine-tuned using different strategies across three tasks in our study. By default, the DDPM sampling (Algorithm 4) is used to generate samples for evaluation.

| Method | JS-PwD ($\downarrow$) | JS-TIC ($\downarrow$) | JS-RG ($\downarrow$) | Validity ($\uparrow$) | TM-ens ($\uparrow$) | RMSD-ens ($\downarrow$) |
|---|---|---|---|---|---|---|
| ESMDiff (DDPM) | 0.372 | 0.421 | 0.440 | **0.940** | 0.843 | 1.437 |
| w/o time condition | 0.385 | 0.421 | 0.456 | 0.930 | 0.849 | 1.454 |
| w/ sequence mask | 0.387 | 0.414 | 0.461 | **0.940** | 0.849 | 1.423 |
| w/ sequence dropout | **0.358** | **0.408** | **0.431** | 0.910 | 0.846 | 1.407 |
| w/ sequence prediction | 0.366 | 0.442 | 0.474 | 0.860 | **0.853** | **1.389** |

| Method | Apo/holo | | | Fold-switch | | |
|---|---|---|---|---|---|---|
| | ResFlex $r$ (gl.) | ResFlex $r$ (pt.) | TM-ens | ResFlex $r$ (gl.) | ResFlex $r$ (pt.) | TM-ens |
| ESMDiff (DDPM) | **0.420** | **0.489** / 0.515 | 0.838 / 0.877 | **0.402** | **0.341** / 0.288 | 0.626 / 0.685 |
| w/o time condition | 0.419 | 0.479 / 0.487 | **0.847** / 0.883 | 0.346 | 0.340 / **0.357** | **0.633** / 0.685 |
| w/ sequence mask | 0.371 | 0.474 / 0.491 | 0.843 / **0.885** | 0.381 | 0.336 / 0.329 | 0.624 / **0.687** |
| w/ sequence dropout | 0.345 | 0.460 / 0.483 | 0.833 / 0.858 | 0.373 | 0.323 / 0.308 | 0.597 / 0.665 |
| w/ sequence prediction | 0.419 | 0.456 / 0.445 | 0.828 / 0.860 | 0.360 | 0.324 / 0.341 | 0.598 / 0.626 |

| Method | Pairwise distance | Radius of gyration | Contact map |
|---|---|---|---|
| ESMDiff (DDPM) | **7.010** / 4.665 | **4.438** / 1.974 | **0.354** / **0.284** |
| w/o time condition | 7.399 / 4.855 | 4.854 / **1.838** | 0.366 / 0.283 |
| w/ sequence mask | 7.027 / **4.654** | 4.523 / 2.221 | 0.363 / 0.293 |
| w/ sequence dropout | 7.140 / 4.904 | 4.629 / 2.502 | 0.369 / 0.303 |
| w/ sequence prediction | 8.122 / 4.764 | 5.414 / 2.422 | 0.433 / 0.361 |

et al. (2024a), the authors established a comprehensive benchmark and dataset split to evaluate how well different conformation generation methods capture various statistical properties when learning from MD ensembles. As shown in Table S6, ESM3-based models, including both pre-trained ESM and ESMDiff, struggled to accurately reconstruct the MD ensembles from the ATLAS test set. This limitation may stem from ATLAS's relatively short 100ns simulation timescale, which rarely captures slower, substantial conformational changes (as noted by Jing et al. (2024a)). Such brief timescales particularly challenge tokenized structure and language modeling approaches, as the categorical nature of structure tokens makes it difficult to represent subtle structural changes in token relationships. Despite the efficiency of SLMs, developing better methods to leverage these SLMs for learning from MD simulation ensembles represents a promising direction for future research.

Table S6: Statistical metrics on MD ensembles of ATLAS test set (Jing et al., 2024a).

| Metrics | AlphaFlow-MD | ESM3 (ID) | ESMDiff (ID) |
|---|---|---|---|
| Pairwise RMSD r | 0.48 | 0.08 | 0.18 |
| Global RMSF r | 0.60 | 0.19 | 0.49 |
| Per target RMSF r | 0.85 | 0.67 | 0.68 |
| RMWD | 2.61 | 7.27 | 7.48 |
| RMWD trans | 2.28 | 5.22 | 5.18 |
| RMWD var | 1.30 | 4.35 | 3.37 |
| MD PCA W2 | 1.52 | 2.06 | 2.29 |
| Joint PCA W2 | 2.25 | 5.97 | 6.32 |
| PC sim 0.5 % | 44 | 22 | 23 |
| Weak contacts J | 0.62 | 0.45 | 0.52 |
| Transient contacts J | 0.41 | 0.26 | 0.26 |
| Exposed residue J | 0.50 | - | - |
| Exposed MI matrix rho | 0.25 | - | - |

# F STRUCTURE AUTO-ENCODERS

## F.1 QUANTIZED REPRESENTATION OF STRUCTURES

Tokenizing protein structures can differ from the cases of image pixels (Van Den Oord et al., 2017; Ramesh et al., 2021). The encoded structure should not only provide informative representations for accurate reconstruction but also ensure a well-structured latent space that enables efficient language modeling. In order to build better structure language models, we discuss several rules that ideal structure quantization methods should follow.

**Disentangled quantization.** Firstly, we ask the disentangled property of the latent $z^{\parallel}$. In specific, each position of the codes $z_i \in |V|$ has only a limited receptive field of the local structure and the interaction between different codes should be minimized. In other words, specific token $z_i$ is expected to correspond to a specific pattern of local structures, where small noise from inputs can be redacted or inferred from other tokens. Disentangled representation can enable efficient language modeling and the interpretability of the latent space. Formally, we assume *conditional independence* of $z_i$ given input structure, i.e. $q(z|x) \equiv q(z_1, z_2, \ldots, z_N|x) = q(z_1|x)q(z_2|x)\ldots q(z_N|x)$. [**]

**Roto-translation invariant encoding.** Roto-translation invariance (RT-Inv) of a function (mapping) indicates that applying (global) affine transformations $T \circ x = R \circ x + t$, composed of 3D rotation matrix $R \in \mathbb{R}^{3 \times 3}$ and translation vector $t \in \mathbb{R}^3$, to the input (of Euclidean space) should have no effects on the final output value. Formally, the encoder $q(z|x) : \mathcal{X} \to \mathcal{Z}$ with RT-Inv property can be described as: $q(z|T \circ x) = q(z|R \circ x + t) = q(z|x), \forall T$. RT-Inv property is essential for effective structure quantization since encoding global rotation or translation into representation can make unnecessary assumption for network architectures. It is also non-trivial to encode orientation or translation in the categorical codes $z$.

**Complete coverage of conformation space.** The (approximately) complete coverage property of quantized representation guarantees successfully reconstruction between $x$ and $\tilde{x}$. Good latent space $\mathcal{Z}$ should be properly configured according to the distribution of structure characteristics. To achieve good reconstruction, latent codes $z$ should be able to catch and distinguish various subtle structure patterns from the input space $\mathcal{X}$ for high-quality decoding. On the other hand, excessively large vocabulary can cause a sparse coding space and unnecessarily encode the noises in structure, which can thus lead to memorization (Arpit et al., 2017) and make the generalization difficult.

## F.2 RECONSTRUCTION ON TEST DATASETS

To investigate whether the structure tokens by the dVAE (Hayes et al., 2024) can encode different conformations in a distinguishable way, we evaluate the reconstruction accuracy of the encode-decoder as a preliminary study via the root-mean-square-deviation (RMSD) on the benchmarking test datasets used in this study. The results are presented as histogram as shown in Fig S3. Note that a non-negligible ratio of the fold-switching Chakravarty & Porter (2022) and IDPs (Lazar et al., 2021) targets for which the encoder-decoder cannot reconstruct the input protein structure very well (RMSD $> 0.5$Å) passing through the quantization process.

Furthermore, we plot the histogram for pairing structures of each target in the Apo/holo (Saldaño et al., 2022) and fold-switching Chakravarty & Porter (2022) for a closer investigation. As shown in Fig. S4 (a) and (b) respectively, the dVAE can have a better accuracy for the folds in the Apo/holo dataset than those of fold-switching, which may explain the relatively inferior performance of SLMs on the fold-switching test data.

---

[∥]To avoid ambiguity, we confine the meaning of "disentangle" to be local-to-global instead of semantic-level description of the structure (eg. double-helix). Thus, the latent $z_i$ should have restricted receptive field.

[**]Note that conditional independence of $z$ given $x$ does not necessarily imply such relation given $c$, though these two variables, as protein sequence and structure, possess high mutual dependence.

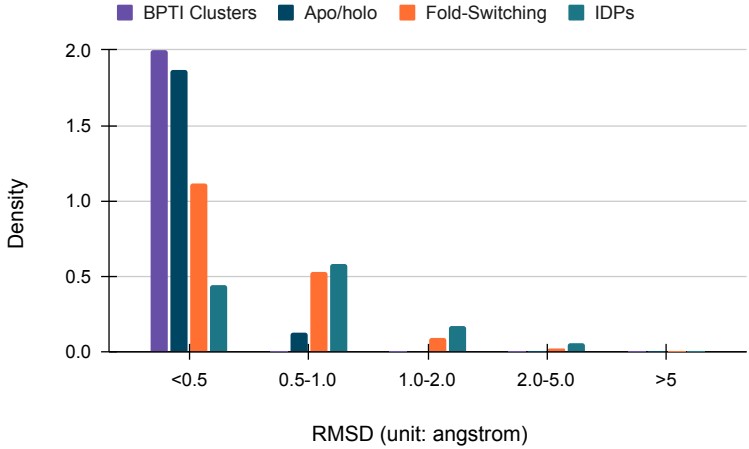

Figure S3: Histogram of RMSD of the targets from each participating test set used in this study. Frequencies has been normalized as density due to the unequal sizes of different datasets.

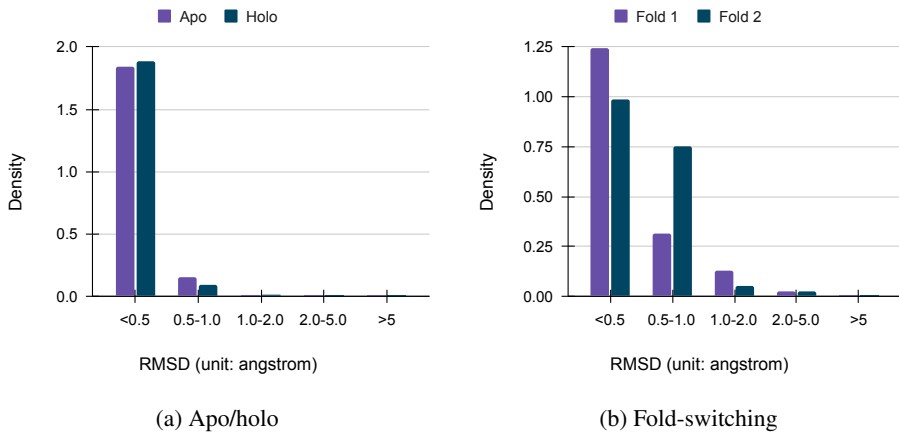

(a) Apo/holo

(b) Fold-switching

Figure S4: Histogram of RMSD of the targets in the (a) Apo/holo pairs; (b) Fold-switching pairs.

## G EXTENDED METHODOLOGY DISCUSSION

### G.1 REVISITING DISCRETE DIFFUSION AS DISTRIBUTION INTERPOLATION

The discrete diffusion models (Austin et al., 2021; Lou et al., 2023; Sun et al., 2022; Campbell et al., 2022; Zheng et al., 2023a) can be generally defined by a sequential process of progressive noisy variables $z_t \in V$ from the categorical variable $z_0 \in V$. Denote the one-hot (row) vector of $z_t$ as $\boldsymbol{z}_t \in \{0,1\}^{|V|}$, in the discrete-time case (Austin et al., 2021), the forward marginal probability of $\boldsymbol{z}_t$ at time $t$ has the following form as a composition of Markov kernel defined by $\boldsymbol{Q}_t$ $(t = 1, 2\ldots, T)$:

$$q(\boldsymbol{z}_t|\boldsymbol{z}_0) = \mathrm{Cat}\left(\boldsymbol{z}_t; \boldsymbol{z}_0\bar{\boldsymbol{Q}}_t\right) \triangleq \mathrm{Cat}\left(\boldsymbol{z}_t; \boldsymbol{z}_0\boldsymbol{Q}_1 \cdot \cdots \cdot \boldsymbol{Q}_t\right), \tag{7}$$

where $\boldsymbol{Q}_t$ indicates the transition probability matrix for time $t$ represented by $[\boldsymbol{Q}_t]_{ij} = q(z_t = j|z_{t-1} = i)$, and $\mathrm{Cat}(\cdot; \boldsymbol{p}), \boldsymbol{p} \in \Delta^{|V|}$ indicates the categorical distribution with probability and $\Delta^{|V|}$ is the $|V|$-simplex. Eq. 7 also induce the form of the marginal distribution for $\forall t > s$ is $q(\boldsymbol{z}_t|\boldsymbol{z}_s) = \mathrm{Cat}\left(\boldsymbol{z}_t; \boldsymbol{z}_s\bar{\boldsymbol{Q}}_{t|s}\right) \triangleq \mathrm{Cat}\left(\boldsymbol{z}_t; \boldsymbol{z}_s\boldsymbol{Q}_{s+1} \cdot \cdots \cdot \boldsymbol{Q}_t\right)$. Correspondingly, the posterior $q(\boldsymbol{z}_s|\boldsymbol{z}_t, \boldsymbol{z}_0)$ can be obtained by the reverse process (Austin et al., 2021):

$$q(\boldsymbol{z}_s|\boldsymbol{z}_t, \boldsymbol{z}_0) = \frac{q(\boldsymbol{z}_t|\boldsymbol{z}_s, \boldsymbol{z}_0)q(\boldsymbol{z}_s|\boldsymbol{z}_0)}{q(\boldsymbol{z}_t|\boldsymbol{z}_0)} = \mathrm{Cat}\left(\boldsymbol{z}_s; \frac{\boldsymbol{z}_t\boldsymbol{Q}_{t|s}^\top \odot \boldsymbol{z}_0\bar{\boldsymbol{Q}}_s}{\boldsymbol{z}_0\bar{\boldsymbol{Q}}_t\boldsymbol{z}_t^\top}\right), \forall\, s < t. \tag{8}$$

Both Zhao et al. (2024) and Shi et al. (2024) discuss how the discrete-time diffusion process can be generalized to the time domain $t \in [0, 1]$, akin to the diffusion over continuous space (Song et al., 2020), by demonstrating the continuous-time limit as $T \to \infty$. Notably, when the stationary distribution is explicitly specified (denoted as $\boldsymbol{p} \in \Delta^{|V|}$), we can choose a *state-independent* transition kernel in the simple form: $\boldsymbol{Q}_{t|s} \triangleq \left[\alpha(s)^{-1}\alpha(t)\boldsymbol{I} + (1 - \alpha(s)^{-1}\alpha(t))\mathbf{1}^\top\boldsymbol{p}\right]$, thus simplifying the continuous-time forward marginal to:

$$q(\boldsymbol{z}_t|\boldsymbol{z}_s) = \mathrm{Cat}\left(\boldsymbol{z}_t; \frac{\alpha(t)}{\alpha(s)}\boldsymbol{z}_s + (1 - \frac{\alpha(t)}{\alpha(s)})\boldsymbol{p}\right), \forall\, 0 \le s < t < 1, \tag{9}$$

where $\alpha(t) \in [0, 1)$ is a strictly monotone decreasing function with $\alpha_0 = 1$ and $\alpha_1 \to 0$. The equation above demonstrates that the discrete diffusion, when defined with an explicit stationary distribution, can be viewed as an interpolation between two categorical distributions controlled by $\alpha(t)$. According to Eq. 8, the reverse process of diffusion defined in Eq. 9 takes the following form for the posterior distribution, where $0 \le s < t < 1$:

$$q(\boldsymbol{z}_s|\boldsymbol{z}_t, \boldsymbol{z}_0) = \mathrm{Cat}\left(\boldsymbol{z}_s; \frac{[\mu(t,s)\boldsymbol{z}_t + (1 - \mu(t,s))\lambda_{\boldsymbol{z}_t}(\boldsymbol{p})\mathbf{1}] \odot [\alpha(s)\boldsymbol{z}_0 + (1 - \alpha(s))\boldsymbol{p}]}{\alpha(t)\lambda_{\boldsymbol{z}_t}(\boldsymbol{z}_0) + (1 - \alpha(t))\lambda_{\boldsymbol{z}_t}(\boldsymbol{p})}\right), \tag{10}$$

where $\mu(t,s) \triangleq \alpha(s)^{-1}\alpha(t) > 0$ and indicator function $\lambda_{\boldsymbol{z}_t}(\cdot) \triangleq \langle\boldsymbol{z}_t, \cdot\rangle$ for concision. Readers are referred to Appendix G.2.2 for more details on deriving the Eq. 9 and 10.

Unlike open-ended text generation, protein conformation generation is well-defined within discrete diffusion models, as it conditions on the input amino acid sequence, allowing each output token to correspond uniquely to a position in the input and thus enjoy a fixed-length context window[††].

### G.2 PROOFS AND DERIVATIONS

#### G.2.1 DERIVATION OF THE EVIDENCE LOWER BOUND IN EQUATION 1

We want to prove the inequality of ELBO of the log-likelihood over structure tokens $\boldsymbol{z}$ conditioned on input $\boldsymbol{c}$, which is:

$$\log p_{\theta,\psi}(\boldsymbol{x}|\boldsymbol{c}) \ge \mathbb{E}_{q_\psi(\boldsymbol{z}|\boldsymbol{x})}\left[\log p_\phi(\boldsymbol{x}|\boldsymbol{c}, \boldsymbol{z}) - D_{\mathrm{KL}}(q_\psi(\boldsymbol{z}|\boldsymbol{x})\|p_\theta(\boldsymbol{z}|\boldsymbol{c}))\right]$$

*Proof:* Consider the log likelihood of structure $\boldsymbol{x}$ and sequence $\boldsymbol{c}$, which is parameterized by parameters $\theta, \psi$:

$$\log p_{\theta,\psi}(\boldsymbol{x}|\boldsymbol{c}) = \log \int p_{\theta,\psi}(\boldsymbol{x}, \boldsymbol{z}|\boldsymbol{c})\, d\boldsymbol{z}$$

---

[††]This indicates that $\boldsymbol{c}_{[i]}$ and $\boldsymbol{z}_{[i]}$ are aligned at the same position index $i$.

where $\boldsymbol{z}$ is introduced as a latent variable, and $p_{\theta,\psi}(\boldsymbol{x},\boldsymbol{z}|\boldsymbol{c})$ is the joint distribution over $\boldsymbol{x}$ and $\boldsymbol{z}$. Write the variational distribution $q_\psi(\boldsymbol{z}|\boldsymbol{x})$ as an approximation to the true posterior over the latent variables $\boldsymbol{z}$ conditioned on $\boldsymbol{x}$, yielding the following log-likelihood:

$$\log p_{\theta,\psi}(\boldsymbol{x}|\boldsymbol{c}) = \log \int \frac{p_{\theta,\psi}(\boldsymbol{x},\boldsymbol{z}|\boldsymbol{c})}{q_\psi(\boldsymbol{z}|\boldsymbol{x})} q_\psi(\boldsymbol{z}|\boldsymbol{x})\, dz,$$

Here we assume positive conditional distribution of $q_\psi(\boldsymbol{c},\boldsymbol{z}|\boldsymbol{x})$. By applying Jensen's inequality to the logarithm, we obtain the following inequality:

$$\log p_{\theta,\psi}(\boldsymbol{x}|\boldsymbol{c}) \geq \mathbb{E}_{q_\psi(\boldsymbol{z}|\boldsymbol{x})}\left[\log \frac{p_{\theta,\psi}(\boldsymbol{x},\boldsymbol{z}|\boldsymbol{c})}{q_\psi(\boldsymbol{z}|\boldsymbol{x})}\right]$$

We can simplify the terms inside the expectation and separates the R.H.S. into two expectations:

$$\mathbb{E}_{q_\psi(\boldsymbol{z}|\boldsymbol{x})}\left[\log \frac{p_{\theta,\psi}(\boldsymbol{x},\boldsymbol{z}|\boldsymbol{c})}{q_\psi(\boldsymbol{z}|\boldsymbol{x})}\right] = \mathbb{E}_{q_\psi(\boldsymbol{z}|\boldsymbol{x})}\left[\log p_\phi(\boldsymbol{x}|\boldsymbol{c},\boldsymbol{z})\right] + \mathbb{E}_{q_\psi(\boldsymbol{z}|\boldsymbol{x})}\left[\log \frac{p_\theta(\boldsymbol{z}|\boldsymbol{c})}{q_\psi(\boldsymbol{z}|\boldsymbol{x})}\right]$$

The second term is the negative Kullback-Leibler (KL) divergence, and thus:

$$\mathbb{E}_{q_\psi(\boldsymbol{z}|\boldsymbol{x})}\left[\log p_\phi(\boldsymbol{x}|\boldsymbol{c},\boldsymbol{z})\right] - D_{\mathrm{KL}}(q_\psi(\boldsymbol{z}|\boldsymbol{x})\|p_\theta(\boldsymbol{z}|\boldsymbol{c}))$$

Since the latent tokens $\boldsymbol{z}$ encode the protein structures, we assume that $\boldsymbol{x}$ *is conditionally independent of $\boldsymbol{c}$ given $\boldsymbol{z}$*. Thus, we obtain:

$$\log p_{\theta,\psi}(\boldsymbol{x}|\boldsymbol{c}) \geq \mathbb{E}_{q_\psi(\boldsymbol{z}|\boldsymbol{x})}\left[\log p_\phi(\boldsymbol{x}|\boldsymbol{c},\boldsymbol{z}) - D_{\mathrm{KL}}(q_\psi(\boldsymbol{z}|\boldsymbol{x})\|p_\theta(\boldsymbol{z}|\boldsymbol{c}))\right]$$

### G.2.2 DERIVATIONS OF MARGINAL AND POSTERIOR IN G.1

Based on the theoretical results in Austin et al. (2021), we derive the marginal and posterior distribution corresponding to the state-independent transition kernel defined in as $\boldsymbol{Q}_{t|s} \triangleq \left[\alpha(s)^{-1}\alpha(t)\boldsymbol{I} + (1 - \alpha(s)^{-1}\alpha(t))\mathbf{1}^\top \boldsymbol{p}\right]$, with $\alpha(t) \in [0,1]$ being some strictly monotone decreasing function s.t. $\alpha_0 = 1$ and $\alpha_1 \to 0$.

**Forward Marginal.** For $q(\boldsymbol{z}_t|\boldsymbol{z}_s)$, we plug in the transition matrix into the definition of forward marginal:

$$
\begin{aligned}
q(\boldsymbol{z}_t|\boldsymbol{z}_s) &= \mathrm{Cat}\left(\boldsymbol{z}_t; \boldsymbol{z}_s \boldsymbol{Q}_{t|s}\right) \\
&= \mathrm{Cat}\left(\boldsymbol{z}_t; \boldsymbol{z}_s\left[\alpha(s)^{-1}\alpha(t)\boldsymbol{I} + (1 - \alpha(s)^{-1}\alpha(t))\mathbf{1}^\top \boldsymbol{p}\right]\right) \\
&= \mathrm{Cat}\left(\boldsymbol{z}_t; \frac{\alpha(t)}{\alpha(s)}\boldsymbol{z}_s + (1 - \frac{\alpha(t)}{\alpha(s)})\boldsymbol{z}_s(\mathbf{1}^\top \boldsymbol{p})\right) \\
&= \mathrm{Cat}\left(\boldsymbol{z}_t; \frac{\alpha(t)}{\alpha(s)}\boldsymbol{z}_s + (1 - \frac{\alpha(t)}{\alpha(s)})(\boldsymbol{z}_s\mathbf{1}^\top)\boldsymbol{p}\right) \\
&= \mathrm{Cat}\left(\boldsymbol{z}_t; \frac{\alpha(t)}{\alpha(s)}\boldsymbol{z}_s + (1 - \frac{\alpha(t)}{\alpha(s)})\boldsymbol{p}\right)
\end{aligned}
\tag{11}
$$

where in the second to last line the associative property is used and $\boldsymbol{z}_s\mathbf{1}^\top = \langle \boldsymbol{z}_s, \mathbf{1}\rangle = 1 \in \mathbb{R}, \forall t$ due to the fact that by definition $\forall s, \boldsymbol{z}_s \in \{0,1\}^N$ is an one-hot (row) vector, i.e. $\langle \boldsymbol{z}_s, \boldsymbol{z}_s\rangle = 1$. Let $\mu(s,t) \triangleq \alpha(s)^{-1}\alpha(t)$ be positive function defined on $[0,1]$, we can write forward marginal as $q(\boldsymbol{z}_t|\boldsymbol{z}_s) = \mathrm{Cat}\left(\boldsymbol{z}_t; \mu(s,t)\boldsymbol{z}_s + (1 - \mu(s,t))\boldsymbol{p}\right)$, which can be viewed as an "interpolation" between categorical distribution $\mathrm{Cat}(\cdot; \boldsymbol{z}_s)$ and stationary prior $\mathrm{Cat}(\cdot; \boldsymbol{p})$.

**Posterior distribution.** Similarly we derive the posterior as follows. Note that by definition: $\bar{Q}_t \equiv Q_{t|0} = \left[\alpha(0)^{-1}\alpha(t)\boldsymbol{I} + (1 - \alpha(0)^{-1}\alpha(t))\boldsymbol{1}^\top \boldsymbol{p}\right] = \alpha(t)\boldsymbol{I} + (1 - \alpha(t))\boldsymbol{1}^\top \boldsymbol{p}, \forall\, t > 0$):

$$q(\boldsymbol{z}_s|\boldsymbol{z}_t, \boldsymbol{z}_0) = \mathrm{Cat}\left(\boldsymbol{z}_s; \frac{\boldsymbol{z}_t \boldsymbol{Q}_{t|s}^\top \odot \boldsymbol{z}_0 \bar{\boldsymbol{Q}}_s}{\boldsymbol{z}_0 \bar{\boldsymbol{Q}}_t \boldsymbol{z}_t^\top}\right),$$

$$= \mathrm{Cat}\left(\boldsymbol{z}_s; \frac{\boldsymbol{z}_t \left[\alpha(s)^{-1}\alpha(t)\boldsymbol{I} + (1 - \alpha(s)^{-1}\alpha(t))\boldsymbol{1}^\top \boldsymbol{p}\right]^\top \odot \boldsymbol{z}_0 \left[\alpha(s)\boldsymbol{I} + (1 - \alpha(s))\boldsymbol{1}^\top \boldsymbol{p}\right]}{\boldsymbol{z}_0 \left[\alpha(t)\boldsymbol{I} + (1 - \alpha(t))\boldsymbol{1}^\top \boldsymbol{p}\right]\boldsymbol{z}_t^\top}\right)$$

$$= \mathrm{Cat}\left(\boldsymbol{z}_s; \frac{\left[\alpha(s)^{-1}\alpha(t)\boldsymbol{z}_t + (1 - \alpha(s)^{-1}\alpha(t))\boldsymbol{z}_t \boldsymbol{p}^\top \boldsymbol{1}\right] \odot \left[\alpha(s)\boldsymbol{z}_0 + (1 - \alpha(s))\boldsymbol{p}\right]}{\left[\alpha(t)\boldsymbol{z}_0 + (1 - \alpha(t))\boldsymbol{p}\right]\boldsymbol{z}_t^\top}\right)$$

$$= \mathrm{Cat}\left(\boldsymbol{z}_s; \frac{\left[\mu(s,t)\boldsymbol{z}_t + (1 - \mu(s,t))\langle \boldsymbol{z}_t, \boldsymbol{p}\rangle \boldsymbol{1}\right] \odot \left[\alpha(s)\boldsymbol{z}_0 + (1 - \alpha(s))\boldsymbol{p}\right]}{\alpha(t)\langle \boldsymbol{z}_0, \boldsymbol{z}_t\rangle + (1 - \alpha(t))\langle \boldsymbol{p}, \boldsymbol{z}_t\rangle}\right). \tag{12}$$

### G.2.3 Derivation of conditional masked diffusion in 4.1

The conditional masked diffusion follows the modeling of forward and reverse processes derived above with the special case that $\boldsymbol{p} = \boldsymbol{p}_M$. Let $\boldsymbol{p}_M \in \{0,1\}^{|\bar{V}|}$ be the one-hot vector for [MASK] token. The forward marginal is directly obtained as

$$q(\boldsymbol{z}_t|\boldsymbol{z}_s) = \mathrm{Cat}\left(\boldsymbol{z}_t; \mu(s,t)\boldsymbol{z}_s + (1 - \mu(s,t))\boldsymbol{p}_M\right), \forall\, 0 \le s < t \le 1. \tag{13}$$

Then we plug this in Eq. 12, which is:

$$q(\boldsymbol{z}_s|\boldsymbol{z}_t, \boldsymbol{z}_0) = \mathrm{Cat}\left(\boldsymbol{z}_s; \frac{\left[\mu(s,t)\boldsymbol{z}_t + (1 - \mu(s,t))\langle \boldsymbol{z}_t, \boldsymbol{p}_M\rangle \boldsymbol{1}\right] \odot \left[\alpha(s)\boldsymbol{z}_0 + (1 - \alpha(s))\boldsymbol{p}_M\right]}{\alpha(t)\langle \boldsymbol{z}_0, \boldsymbol{z}_t\rangle + (1 - \alpha(t))\langle \boldsymbol{p}_M, \boldsymbol{z}_t\rangle}\right). \tag{14}$$

Since multiple inner-product terms appear between one-hot vectors $\boldsymbol{z}_0, \boldsymbol{z}_t, \boldsymbol{p}_M$ which only takes binary value $\in \{0,1\}$, we discuss this by cases and derived the form in Eq. 10:

$\boldsymbol{z}_t = \texttt{[MASK]}$. In this case, we immediately note that $\langle \boldsymbol{z}_t, \boldsymbol{p}_M\rangle = 1$ and $\langle \boldsymbol{z}_t, \boldsymbol{z}_0\rangle = 0$ because $\boldsymbol{z}_0 \in V$ represents the data and thus $\boldsymbol{z}_0 \ne \texttt{[MASK]} = \boldsymbol{z}_t$. Plug in Eq. 14, we have:

$$q(\boldsymbol{z}_s|\boldsymbol{z}_t, \boldsymbol{z}_0) = \mathrm{Cat}\left(\boldsymbol{z}_s; \frac{\left[\mu(s,t)\boldsymbol{p}_M + (1 - \mu(s,t))\boldsymbol{1}\right] \odot \left[\alpha(s)\boldsymbol{z}_0 + (1 - \alpha(s))\boldsymbol{p}_M\right]}{1 - \alpha(t)}\right)$$

$$= \mathrm{Cat}\left(\boldsymbol{z}_s; \frac{\left[\mu(s,t)\boldsymbol{p}_M + (1 - \mu(s,t))\boldsymbol{1}\right] \odot \left[\alpha(s)\boldsymbol{z}_0 + (1 - \alpha(s))\boldsymbol{p}_M\right]}{1 - \alpha(t)}\right)$$

$$= \mathrm{Cat}\left(\boldsymbol{z}_s; \frac{\left[\mu(s,t)\boldsymbol{p}_M + (1 - \mu(s,t))\boldsymbol{1}\right] \odot \left[\alpha(s)\boldsymbol{z}_0 + (1 - \alpha(s))\boldsymbol{p}_M\right]}{1 - \alpha(t)}\right)$$

$$= \mathrm{Cat}\left(\boldsymbol{z}_s; \frac{\left[\mu(s,t)\boldsymbol{p}_M + (1 - \mu(s,t))\boldsymbol{1}\right] \odot \left[\alpha(s)\boldsymbol{z}_0 + (1 - \alpha(s))\boldsymbol{p}_M\right]}{1 - \alpha(t)}\right)$$

$$= \mathrm{Cat}\left(\boldsymbol{z}_s; \frac{\mu(s,t)\alpha(s)\boldsymbol{0} + \mu(s,t)(1 - \alpha(s))\boldsymbol{p}_M + (1 - \mu(s,t))\alpha(s)\boldsymbol{z}_0 + (1 - \mu(s,t))(1 - \alpha(s))\boldsymbol{p}_M}{1 - \alpha(t)}\right)$$

$$= \mathrm{Cat}\left(\boldsymbol{z}_s; \frac{\alpha(s) - \alpha(t)}{1 - \alpha(t)}\boldsymbol{z}_0 + \frac{1 - \alpha(s)}{1 - \alpha(t)}\boldsymbol{p}_M\right), \tag{15}$$

where we use the facts that $\boldsymbol{z}_0 \odot \boldsymbol{p}_M = \boldsymbol{0}, \boldsymbol{p}_M \odot \boldsymbol{p}_M = \boldsymbol{p}_M$, and $* \odot \boldsymbol{1} = *$.

$\boldsymbol{z}_t \ne \texttt{[MASK]}$. Correspondingly, this leads to $\langle \boldsymbol{z}_t, \boldsymbol{p}_M\rangle = 0$ and we have the following simplification:

$$q(\boldsymbol{z}_s|\boldsymbol{z}_t, \boldsymbol{z}_0) = \mathrm{Cat}\left(\boldsymbol{z}_s; \frac{\left[\mu(s,t)\boldsymbol{z}_t + (1 - \mu(s,t))\langle \boldsymbol{z}_t, \boldsymbol{z}_0\rangle \boldsymbol{1}\right] \odot \left[\alpha(s)\boldsymbol{z}_0 + (1 - \alpha(s))\boldsymbol{p}_M\right]}{\alpha(t)\langle \boldsymbol{z}_t, \boldsymbol{z}_0\rangle + (1 - \alpha(t))\langle \boldsymbol{z}_t, \boldsymbol{p}_M\rangle}\right)$$

$$= \mathrm{Cat}\left(\boldsymbol{z}_s; \frac{\mu(s,t)\alpha(s)(\boldsymbol{z}_t \odot \boldsymbol{z}_0)}{\alpha(t)\langle \boldsymbol{z}_t, \boldsymbol{z}_0\rangle}\right)$$

$$= \mathrm{Cat}\left(\boldsymbol{z}_s; \frac{\boldsymbol{z}_t \odot \boldsymbol{z}_0}{\langle \boldsymbol{z}_t, \boldsymbol{z}_0\rangle}\right). \tag{16}$$

Here we find that, due to the construction of forward marginal in Eq. 13, $z_t \in \{[\text{MASK}], z_0\}$ to guarantee the denominator $\alpha(t)\langle z_t, z_0 \rangle + (1 - \alpha(t))\langle z_t, p_M \rangle$ to be positive. Such that $z_t \neq [\text{MASK}]$ will imply $\langle z_t, z_0 \rangle = 1$ and thus $q(z_s|z_t, z_0)$ becomes $\text{Cat}(z_s; z_t)$ which has probability mass $P(z = z_t) = 1$ and zero elsewhere.

To combine these two cases together, we introduce the indicator function $\lambda_M(\cdot) \triangleq \langle p_M, \cdot \rangle$ and let $\beta(s,t) \triangleq \frac{1-\alpha(s)}{1-\alpha(t)} > 0$, Eq. 2 can be obtained by plugging in the coefficients to Eq. 14:

$$q(z_s|z_t, z_0) = \text{Cat}\left(z_s; [\beta(s,t) + (1 - \lambda_M(z_t))(1 - \beta(s,t))]z_t + \lambda_M(z_t)(1 - \beta(s,t))z_0\right). \tag{17}$$

### G.2.4 LEARNING OBJECTIVE FOR CONDITIONAL MASKED DIFFUSION

Based on the pre-defined masked diffusion process, now we derive the training objective as in Eq. 3. The task we are interested in involves a key step of sampling structure tokens $z_0$ based on the amino-acid sequence condition $c$. We start with discrete-time denoising diffusion objective according to Ho et al. (2020), which is the ELBO of marginal likelihood of $z_0$ conditioned on $c$ (let $t_i \triangleq \max(\epsilon, i/T), i = 0, 1, \dots, T$):

$$\log p(z_0|c) \geq \mathbb{E}_{q(z_{t_0}|z_0)}[\log p_\theta(z_0|z_{t_0}, c)] - D_{\text{KL}}(q(z_T|z_0)\|p_\theta(z_T|c))$$
$$- \sum_{i=2}^{T} D_{\text{KL}}(q(z_{t_{i-1}}|z_{t_i}, z_0)\|p_\theta(z_{t_{i-1}}|z_{t_i}, c)), \tag{18}$$

where each term can be respectively viewed as *reconstruction*, *prior* and (discrete-time) *diffusion* loss. The definition above is general and still valid regardless of continuous or discrete data. Due to the special construction of conditional masked diffusion in Eq. 9 and 10, we find the cancellations of the first two terms because: (1) For reconstruction loss, the interpolation diffusion process is based on two one-hot vectors $z_0$ and [MASK] which assigns zero probability to all other values. That indicates $z_0 \equiv z_{t_0}$, or there is no "decoding process" mapping from the infinite small time step to original data, i.e. $\lim_{t \to 0} \alpha(t) = 1$. Thus $\log p_\theta(z_0|z_{t_0}, c)] = \log q(z_0|z_{t_0}, u_\theta(z_{t_0}, t_0, c))] \to 0$ as $t_0 \to 0$; (2) For the prior loss, both $q(z_T|z_0)$ and $p_\theta(z_T|c)$ are designed to be $\text{Cat}(z_T|p_M)$ as the stationary distribution with all probability mass on the [MASK] token. Thus, $D_{\text{KL}}(q(z_T|z_0)\|p_\theta(z_T|c)) = D_{\text{KL}}(\text{Cat}(z_T|p_M)\|\text{Cat}(z_T|p_M)) \equiv 0$.

The ELBO in Eq. 18 is simplified to contain only the multi-step diffusion loss, by parameterization of the posterior distribution:

$$\text{ELBO} = -\sum_{i=2}^{T} D_{\text{KL}}(q(z_{t_{i-1}}|z_{t_i}, z_0)\|p_\theta(z_{t_{i-1}}|z_{t_i}, c)),$$
$$= -\sum_{i=2}^{T} D_{\text{KL}}(q(z_{t_{i-1}}|z_{t_i}, z_0)\|q(z_{i-1}|z_{t_i}, u_\theta(z_{t_i}, t_i, c))),$$
$$= -\sum_{i=2}^{T} \sum_{j=1}^{|\bar{V}|} q(z_{t_{i-1}} = e_j|z_{t_i}, z_0) \log \frac{q(z_{t_{i-1}} = e_j|z_{t_i}, z_0)}{q(z_{t_{i-1}} = e_j|z_{t_i}, u_\theta(z_{t_i}, t_i, c)))}, \tag{19}$$

where $e_j \in \{0, 1\}^{|\bar{V}|}$ is the one-hot basis vector with $j$-th element being non-zero. Note that in Eq. 16, we show that if $z_t \neq p_M$, the posterior $q(\cdot|z_t, z_0)$ becomes $z_0$-independent, which imply

$D_{\text{KL}} = 0$. When $z_t = p_M$, according to Eq. 15, we have:

$$
\begin{aligned}
\text{ELBO} &= -\sum_{i=2}^{T} \sum_{j=1}^{|\bar{V}|} q(z_{t_{i-1}} = e_j | z_{t_i}, z_0) \log \frac{q(z_{t_{i-1}} = e_j | z_{t_i}, z_0)}{q(z_{t_{i-1}} = e_j | z_{t_i}, u_\theta(z_{t_i}, t_i, c)))} \\
&= -\sum_{i=2}^{T} \sum_{j=1}^{|V|} (1 - \beta(t_{i-1}, t_i)) \langle z_0, e_j \rangle \log \frac{\langle z_0, e_j \rangle}{\langle u_\theta(z_{t_i}, t_i, c), e_j \rangle} \\
&= -\sum_{i=2}^{T} (1 - \beta(t_{i-1}, t_i)) \sum_{j=1}^{|V|} [\langle z_0, e_j \rangle \log \langle z_0, e_j \rangle - \langle z_0, e_j \rangle \log \langle u_\theta(z_{t_i}, t_i, c), e_j \rangle] \\
&= \sum_{i=2}^{T} (1 - \beta(t_{i-1}, t_i)) \sum_{j=1}^{|V|} \langle z_0, e_j \rangle \log \langle u_\theta(z_{t_i}, t_i, c), e_j \rangle \\
&= \sum_{i=2}^{T} (1 - \beta(t_{i-1}, t_i)) \, \mathbb{E}_{q(z_{t_i} | z_0)} \left[ \langle \log u_\theta(z_{t_i}, t_i, c), z_0 \rangle \right].
\end{aligned} \tag{20}
$$

Then we push $T \to \infty$ and obtain the limit of continuous-time ELBO:

$$
\begin{aligned}
\text{ELBO}_\infty &= \lim_{T \to \infty} \sum_{i=2}^{T} (1 - \beta(t_{i-1}, t_i)) \, \mathbb{E}_{q(z_{t_i} | z_0)} \left[ \langle z_0, \log u_\theta(z_{t_i}, t_i, c) \rangle \right] \\
&= \lim_{T \to \infty} \sum_{i=2}^{T} \frac{\alpha(t_{i-1}) - \alpha(t_i)}{1 - \alpha(t_i)} \, \mathbb{E}_{q(z_{t_i} | z_0)} \left[ \langle z_0, \log u_\theta(z_{t_i}, t_i, c) \rangle \right] \\
&= -\lim_{T \to \infty, \, i/T \to 0} \sum_{i=2}^{T} \frac{[\alpha(t_i) - \alpha(t_{i-1})](t_i - t_{i-1})}{[1 - \alpha(t_i)](t_i - t_{i-1})} \, \mathbb{E}_{q(z_{t_i} | z_0)} \left[ \langle z_0, \log u_\theta(z_{t_i}, t_i, c) \rangle \right] \\
&= -\int_{t \in (0,1)} \frac{1}{1 - \alpha(t)} \frac{\partial \alpha(t)}{\partial t} \, \mathbb{E}_{q(z_t | z_0)} \left[ \langle z_0, \log u_\theta(z_t, t, c) \rangle \right] \mathrm{dt}.
\end{aligned} \tag{21}
$$

In practice, the integral above can be replaced by the uniform sampling $t \sim U[0, 1]$ as Monte Carlo estimation during training. Note that the $\text{ELBO}_\infty$ above is conditioned on $z_t = p_M$ (ELBO=0 otherwise), such that we introduce the indicator function $\lambda_M(z_t)$ and obtain the Eq. 3.

## H    STRUCTURE GALLERY

This section showcases various structural ensembles generated by our model, ESMDiff (DDPM sampling), across different protein scenarios. The visualizations include comparisons between sampled ensembles and reference structure(s) or ground truth ensembles, providing an insight into the model's ability to capture diverse conformations and structural flexibility in different types of proteins.

### H.1    PROTEIN BPTI

Figure S5 displays the best-matched samples from our modeled ensemble ($N = 100$) for each kinetic cluster (in total 5) of BPTI published in Shaw et al. (2010). The superimposed visualization compares the modeled conformations (solid rainbow color) with the reference structure (rendered transparent) after structural alignments.

### H.2    CONFORMATION CHANGE PAIRS

Figure S6 presents the sampled ensemble in khaki, overlaid on the ground truth structures of conformation-changing pairs from the Apo/Holo dataset, shown in transparent rainbow colors.

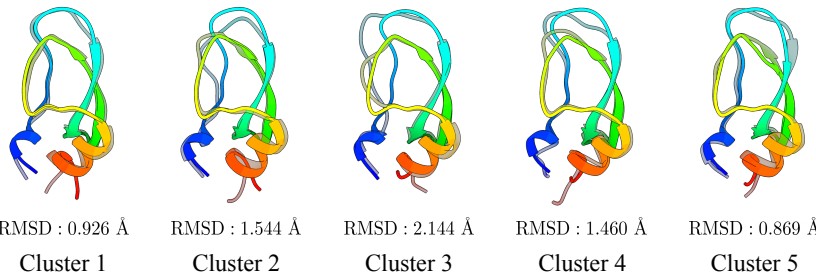

| RMSD : 0.926 Å | RMSD : 1.544 Å | RMSD : 2.144 Å | RMSD : 1.460 Å | RMSD : 0.869 Å |
| --- | --- | --- | --- | --- |
| Cluster 1 | Cluster 2 | Cluster 3 | Cluster 4 | Cluster 5 |

Figure S5: The visualization shows the best matched samples for each kinetic cluster of BPTI from the modeled ensemble ($N = 100$) by ESMDiff. The reference structure is rendered transparent.

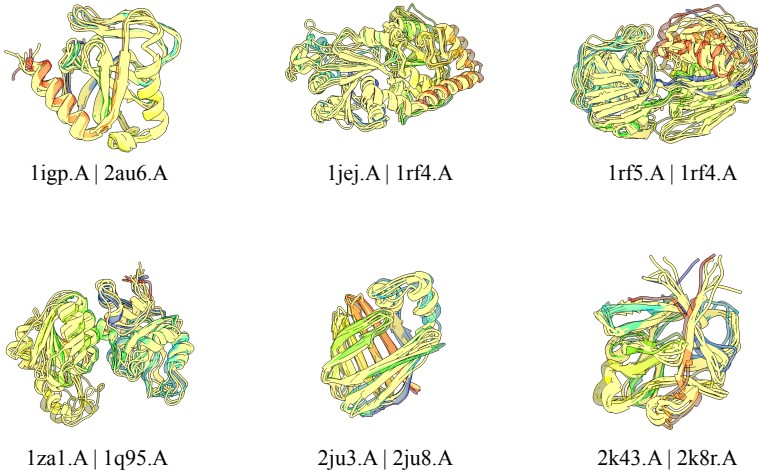

| 1igp.A | 2au6.A | 1jej.A | 1rf4.A | 1rf5.A | 1rf4.A |
| --- | --- | --- |

| 1za1.A | 1q95.A | 2ju3.A | 2ju8.A | 2k43.A | 2k8r.A |
| --- | --- | --- |

Figure S6: Sampled ensemble (colored in khaki) superimposed with the ground truth conformation changing pairs in Apo/Holo dataset(represented in transparent rainbow colors)

## H.3 INTRINSICALLY DISORDERED PROTEIN

The final subsection focuses on intrinsically disordered proteins (IDPs), which are characterized by their flexible, non-fixed structures. Figure S7 demonstrates the sampled ensemble in khaki, superimposed on the ground truth ensemble of IDPs, rendered in transparent rainbow colors. Additionally, the figure includes a comparison of contact maps to the right, further illustrating the model's capacity to reproduce the flexible and dynamic nature of IDPs. Among the examples, we also show a failed case for `PED00433`, where the model struggles to enforce structural diversity at both terminal fragments. While the experimental ensemble exhibits significant structural variability, the model-generated ensemble shows reduced diversity, indicating a potential direction for further improvement in modeling terminal regions of the modern structure prediction models.

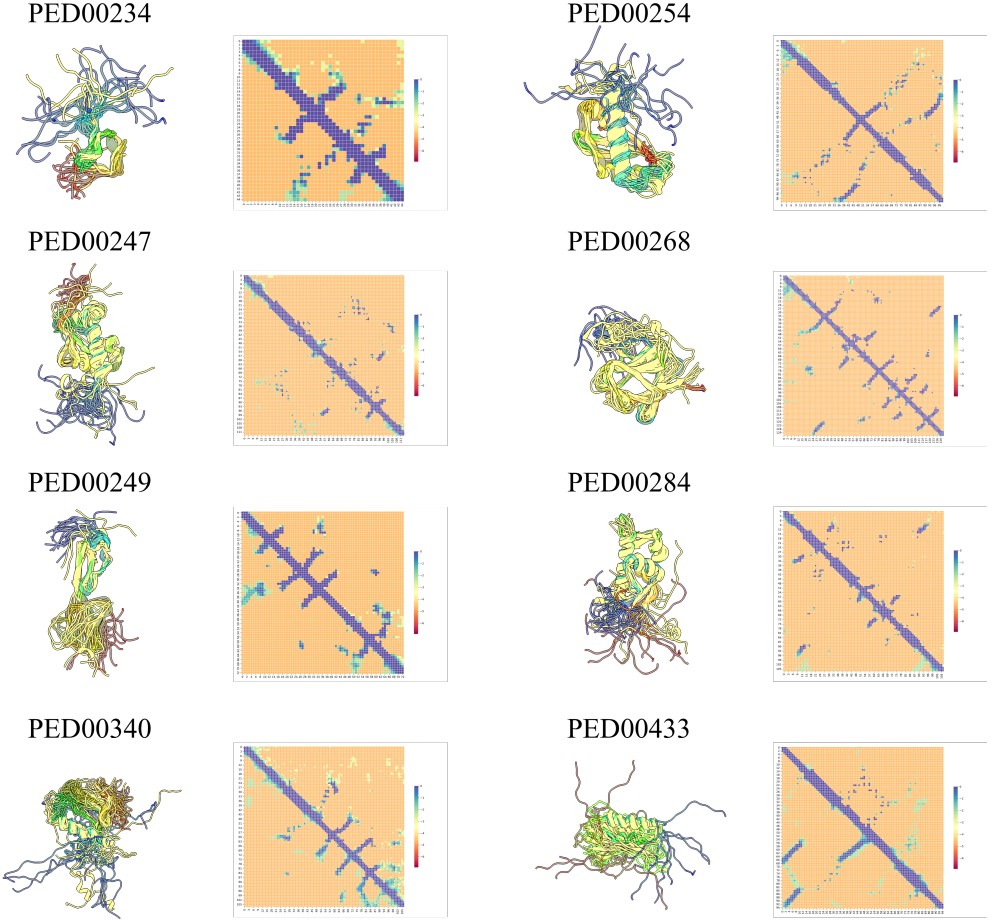

Figure S7: Sampled ensemble (colored in khaki) superimposed with the ground truth ensemble of intrinsically disordered proteins (IDPs) (represented in transparent rainbow colors), accompanied by a contact map comparison on the right.

