# OpenReview forum: "Structure Language Models for Protein Conformation Generation"
_ICLR.cc/2025/Conference — ICLR 2025 Poster_

### Official Review · Reviewer_X2Sg · 2024-10-31

**Soundness:** 3
**Presentation:** 3
**Contribution:** 2
**Rating:** 6
**Confidence:** 2

**Summary:**

This paper aims to generate protein conformation ensembles with Structure Language Modeling (SLM), a method that first tokenizes protein 3D structure through learned autoencoding with a VQVAE, and then generates the 3D structure through language modeling and decoding. Compared to alternative methods such as equivariant diffusion models, the proposed method enjoys a significant efficiency advantage. The proposed method is applied to protein conformation ensemble generation tasks with a diverse set of evaluation metrics.

**Strengths:**

- The proposed method offers an interesting alternative to the task of protein conformation sampling, a task that has so far mostly been tackled with 3D-aware generative models that may suffer from poor computational efficiency. The proposed method naturally leverages pre trained ESM3 model and extends its capablities.

- The experiments validate the efficiency of the proposed method. The performance seems competitive with SOTA structure-based generative models.

**Weaknesses:**

- There are many metrics being benchmarked and it is hard to evaluate the model performance. It seems that the proposed method is quite competitive with SOTA alternatives. However, this reviewer is not fully convinced about that because the results are so scattered around in different tasks and different evaluation metrics, and the proposed method sometimes works better and sometimes works worse compared to baseline methods. Can the authors explain what metrics are the most practically relevant, or propose consolidated metrics for comparing different models? Can the authors include benchmarks that were originally used in the AlphaFlow/ESMFLow paper (Jing et al, 2024), if they are solving the same task?

**Questions:**

Please see "Weaknesses".

---

> ### Author Response · Authors · 2024-11-17
> **Response to reviewer X2Sg - part 1**
>
> Dear reviewer X2Sg,
>
> Thank you very much for the helpful suggestion and question. We want to address them correspondingly as follows:
>
> **(for metric)**:
> - We have carefully selected a relatively diverse range of tasks and used corresponding metrics that require different capabilities to comprehensively evaluate both the pros and cons of our method, instead of only reporting tasks where our method performs better.
> - We also want to highlight that our SLM models are better than other SOTA sequence-based methods across almost all tasks and metrics. This provides enough evidence of the advantages of our SLM methods, as a family of sequence-based methods training using the same PDB data. We also note that MSA-based methods sometimes outperform our method, especially in the case of conformation changing pairs and IDP tasks, while at the cost of being more computationally expensive in practice and strongly dependent on the number of available homologous sequences.
> - We believe the “key metrics” are closely related to the exact application (as well as their available format of “ground truth” to compare). There is no universal definition in the current related study. For example, for IDP researchers, the stable reference structure does not exist, so we ground the evaluation on the discrepancy of distribution or statistics; for fold-switchers such as enzyme/kinase-like target, we can curate the ground truth pairs from the database for evaluation, thus pairwise TM-score can play a part.
> In practice, we state and reason the most relevant metric as follows (those metrics would  be highlighted in the next revision if needed):
>     - for BPTI (Table 1), we prefer JS-PWD (introduced in Str2Str, the symmetric distributional metric based on the most abundant invariant feature vector of structures);
>     - for apo-holo and fold-switch (Table 3), we believe TM-ens (introduced in EigenFold) can truthfully reflect the matching performance for both ground-truth targets;
>     - for IDP (Table 4), we use MAE-PwD because the PwD features contain more abundant information (more dimensions) than the other two.
> - Last, we understand the concern that it is hard to tell which model is superior without a comprehensive metric (ideally we want some metric like FID or Inception score for image generation). That raises the special need for a comprehensive benchmark covering diverse data of protein types and proper metrics in our community.
>
> **(for Atlas data and benchmark)**:
>
> - We are happy to compare with AlphaFlow, which is a very nice relevant work. For AlphaFlow/ESMFlow, we however think the data setting in their work is not quite aligned with our LM model design. This is because the training/evaluation 100ns MD simulation simply produces the rather **limited and local movement** of the system. We note that, AlphaFold, as well as the flow matching variant AlphaFlow, show limited diversity but it can be good at capturing the small, local conformation moving with small Euclidean distance (on average 2 angstroms in the trajectory, see below). The geometric space-based models such as AlphaFlow, are easy to learn the distribution over these highly frequent, “local” distributions, since they are closely neighboring in the euclidean space. \
> - However, in SLM, since we quantize the structure into distinct tokens, the geometric distance and edit distance of tokens are not smoothly correlated. We have examined the Atlas dataset and unfortunately found a surprisingly high (>50% on average) edit distance between conformation in the same trajectory, even though they show on average 1-2 A RMSD. Based on this limitation, it is promising to point out future work that design better architectures capturing the correlation from geometric distance to properly defined latent token distance.
> - As an probing investigation, we firstly reported some trajectory RMSD in mean and max as follows:
>
> |Test Target|Trajectory RMSD-mean|Trajectory RMSD-max|
> |--------------|--------------|--------|
> |6o2v_A|1.345 A|2.619 A|
> |7ead_A|1.568 A|2.296 A|
> |6uof_A|0.803 A|1.237 A|
> |...|...|...|
> |All Test split (mean)|2.042 A| 3.572 A|
>
> Since the targets are general monomeric proteins with hundreds of residues, the average trajectory RMSD ~2 angstrom is so small to reveal significant changes.
>
> (to be continued)

---

> > ### Author Response · Authors · 2024-11-17
> > **Response to reviewer X2Sg - part 2**
> >
> > Secondly, we did some preliminary comparison to further demonstrate the statement above. Since training on the Atlas trajectory data (400k) is over computationally intensive for us during rebuttal, we thus evaluate the training-free ESM3 (in zero-shot) performance on their benchmark compared w/ AlphaFlow for reader’s reference. We use the same sampling configs of 25 steps with 250 samples and evaluate in the bb-only mode because the open-source ESM3 cannot model side-chain atoms (thus the exposed residue is not available).
> >
> > | Metric                        | AlphaFlow-MD | ESM3   |
> > |-------------------------------|--------------|--------|
> > | Pairwise RMSD                 | 2.89         | 7.01   |
> > | Pairwise RMSD r               | 0.48         | 0.08   |
> > | RMSF                          | 1.68         | 3.74   |
> > | Global RMSF r                 | 0.6          | 0.19   |
> > | Per target RMSF r             | 0.85         | 0.67   |
> > | RMWD                          | 2.61         | 7.27   |
> > | RMWD trans                    | 2.28         | 5.22   |
> > | RMWD var                      | 1.3          | 4.35   |
> > | MD PCA W2                     | 1.52         | 2.06   |
> > | Joint PCA W2                  | 2.25         | 5.97   |
> > | PC sim > 0.5 %                | 44           | 21.95  |
> > | Weak contacts J               | 0.62         | 0.45   |
> > | Weak contacts nans            | 0.41         | 0      |
> > | Transient contacts J          | 0.5          | 0.26   |
> > | Transient contacts            | 0.25         | 0      |
> > | Exposed residue J             | 0.50          | NaN    |
> > | Exposed MI matrix rho         | 0.25          | NaN    |
> >
> >
> > From the result, we can tell that the ESM (probably the current version of SLM) is not good at modeling “local” and “short“ because a lot of structures are aggregating in some landscape neighborhood, where explicitly generative modeling in the Euclidean can be more well defined such as AlphaFlow. Following this insightful discussion, we believe that a promising and exciting future work can be designing novel SLM architectures suitable for such MD-based training data that encodes proper inductive bias of the local exploration properties of MD simulation. This may include much better auto-encoders and some sophisticated LM prior models.

---

> > > ### Comment · Reviewer_X2Sg · 2024-11-18
> > >
> > > Thank you for answering my questions. I have raised my rating to 6. It would be good to have a more detailed discussion on the metrics in the next revision.

---

> > > > ### Author Response · Authors · 2024-11-22
> > > > **Response to reviewer X2Sg**
> > > >
> > > > Dear reviewer X2Sg,
> > > >
> > > > We highly appreciate your support and helpful suggestion. We believe our discussion during the review did help improve the quality of our manuscript. We will thus organically incorporate the discussion especially on the metrics into our final revision. Thanks :)
> > > >
> > > > Authors #5043

---

### Official Review · Reviewer_2wMe · 2024-11-02

**Soundness:** 3
**Presentation:** 3
**Contribution:** 3
**Rating:** 8
**Confidence:** 2

**Summary:**

This paper introduces the Structure Language Modeling (SLM) framework for protein conformation generation. The frameworks consist of an auto-encoder and a conditional language model, which can be BERT-like or GPT-like. This method gets a good speedup than previous methods.

**Strengths:**

-  The paper is well motivated.
-  The framework is a meaningful combination of VAE and LM that take the benefits of both.
-  The 20x-100x speed up seems to be promising.

**Weaknesses:**

In results tables, seems all systems (including baselines and SLMs) have different model sizes and pre-training data. It would be better if these details are clearly described in the table to better understand the significance of SLM.

**Questions:**

In Table 2, the best SLM variant in each cluster is all different. Can authors provide more insights about these results?

---

> ### Author Response · Authors · 2024-11-17
> **Response to reviewer 2wMe**
>
> Dear reviewer 2wMe,
>
> Thanks for your supportive comment and time to review our paper! Here is our response to your review.
>
> **Response to weakness:**
>
> Thanks for your good suggestion. Accordingly, we have added the model size of baselines in the Section 5.4 - Runtime analysis to have a good reference of each baseline and discuss the training data used in detail in the Appendix A.5 by carefully reviewing the mentioned baselines. Unlike the purely language models, we think it is not completely fair to directly compare between different types of model (eg., AF2-based inference requires O (N^3) triangle attention computation).  That means smaller model size doesn’t even imply a more “lite” / “efficient” training or inference process.
>
>
> **Response to question:**
>
> For this question, we explain in the following sense: The main difference between SLM variants is their architecture since we keep the training/finetuning data fixed. Since in the latent space, the Euclidean protein structure is quantized into discrete tokens and thus the whole structure is represented by a sequence of tokens z, different model variants can learn in a different way and possibly develop different biases towards a distribution of generated {z}. In this sense, for a specific test case of BPTI, we may imagine the kinetic clusters scattering in the different positions of the space. The generated (conditional) distribution p_<model>(z | sequence) can be different depending on <model> due to the imperfect training, which is because the training does not provide the whole information of such distribution over trajectories during training/finetuning. So the inference is in a sense of “transfer learning”: it aims at learning on static PDB data and “guess” (predict) the possible meta-stable states of BPTI during inference.
>
> Let us know if anything is missing or confusing. We are happy to further discuss and keep improving our manuscript!

---

> ### Author Response · Authors · 2024-11-25
> **Response to reviewer 2wMe**
>
> Dear reviewer 2wMe,
>
> Thank you again for your supportive comments and constructive suggestion! As the author-reviewer discussion deadline is approaching, we want to kindly follow up to see if the response addresses your concerns or if you have any further questions :)
>
> Author #5043

---

### Official Review · Reviewer_yRU8 · 2024-11-02

**Soundness:** 3
**Presentation:** 2
**Contribution:** 2
**Rating:** 6
**Confidence:** 4

**Summary:**

This paper proposes to use protein language models, operating on structure tokens, to model protein conformation ensembles. Several variations on this idea are tried: (1) an encoder-decoder model that translates sequence-tokens to structure-tokens (2) a decoder only model supplied with sequence-token prompts (3) Gibbs sampling with the bidirectional unmasking model ESM3 (4) Gibbs sampling and ancestral sampling with ESMDiff, a version of ESM3 fine-tuned under masked diffusion. These models are evaluated on the BPTI benchmark and conformation pair datasets and compared against several existing methods.

**Strengths:**

* The problem addressed by the authors is important and timely.
* The approach, although based on accessible and popular ideas, has not been explored in the literature and is likely to be of great interest to the community.
* The paper is thorough and has detailed appendices describing training, inference, and evaluation.
* The paper has a diversity of experimental results, including several that appear only in the appendices (?)

**Weaknesses:**

**Novelty**
* The paper's approach is relatively straightforward and is almost an off-the-shelf application of ESM3 or any structural-token protein language model. With that said, I don't think this is a big drawback because the focus of the paper is on the novelty of application and empirical benchmarks.

**Quality**
The paper's evaluations are, however, relatively limited and could be improved in quality and rigor.
* The evaluations lean heavily on the BPTI benchmark, and do not make use of MD datasets like ATLAS and MD-CATH. It would be much more interesting to see how well the model can learn from those datasets and generalize to a larger number of new proteins.
* ESM3, out-of-the-box, can already be used as a sequence->structure generative model with iterative decoding. However, the authors seem to only evaluate ESM3 under Gibbs sampling. Thus a proper comparison with pretrained ESM3 is missing.
* The results themselves are also relatively mixed, with zero-shot ESM3 outperforming ESMDiff in several instances.

**Clarity**
* In all tables, the authors have bolded the best method _among the several methods they propose_, not the best method overall. This gives a very misleading impression and is a major issue that must be fixed.
* At times, the paper has a forced and unnecessary level of formalism. At best, this obscures what the model is actually doing, and at worst leads to misleading or incorrect statements.
    * The authors spend quite a bit of time building up a probabilistic framework for structure based on a standard variational lower bound. However, the authors universally use the ESM3 tokenizer, which is not trained with maximum-likelihood reconstruction. Indeed, "we begin by maximizing the ELBO $\mathcal{L}(\phi, \theta)$ with respect to the encoder $\psi$ and decoder $\phi$" as this is soon undercut by "We start with the pre-trained dVAE established in Hayes et al. (2024) as the structure tokenizer (frozen)." So are $psi$, $\phi$, ever trained, or not? Furthermore, the ESM3 decoder is not conditioned on sequence tokens, unlike the formally defined decoder $p_\phi(x \mid c, z)$.
    * The exposition on general discrete diffusion takes up quite a lot of space, but ultimately the authors use masked diffusion, which could be presented much more clearly and succinctly on its own.
    * Since ESM3 is also trained at all mask levels, it is not immediately clear what is different about the fine-tuning model. As I can currently tell, it is only that the model is explicitly time-conditioned and the unmasking is stochastic instead of a fixed number of tokens unmasked per step. But is this really a significant enough difference to merit a completely different presentation?

To summarize, I like to paper's core idea and would recommend acceptance based on that promise. However, the paper's presentation obfuscates the simplicity of the method, and the evaluations are incompletely and misleadingly presented. It would be preferable for the authors to de-emphasize methodological novelty but focus on a thorough and convincing empirical study of the ability of structural PLMs to sample conformational ensembles.

**Questions:**

See above

---

> ### Author Response · Authors · 2024-11-17
> **Response to reviewer yRU8 - Part 1**
>
> Firstly, we want to kindly thank reviewer yRU8 for the detailed and helpful review. We carefully list our response to your concerns/questions (according to the organized bullet points):
>
> For **novelty**, our methodology mainly aims at presenting “how and why” it makes sense to use presently popular structural autoencoders (eg., [1,2,3]) + latent LMs with empirical studies and efficiency profiling.  For modern AI for Science research, we believe that improving upon well-established foundation models is equally important as developing ad hoc novel models (dominated nowadays). In this sense, we are building up our work and trying to draw more attention to contributions that make the foundation models work well in specific tasks such as conformation generation (providing a recipe). We ponder that a better improvement of our novelty may lie in proposing new invariant architectures in the auto-encoders that are more suitable for latent generative modeling!
>
> For **quality**,
> (1) In MD datasets:
>
> Thanks for the suggestion. We first note that the Atlas dataset is a bit special for their data distribution and not fully suitable for the current application of ESM-like model (we will reason below). We tried performing the fine-tuning for ESM3 on the Atlas training dataset (following Jing et al.),  it seemed too computationally intensive to fit during rebuttal. We instead compare between the AlphaFlow and ESM3 (zero-shot) in the table below. The evaluation is based in the *bb-only* mode because the open-source ESM3 cannot model side-chain atoms (thus the exposed residue is also not available).
>
> | Metric                        | AlphaFlow-MD | ESM3   |
> |-------------------------------|--------------|--------|
> | Pairwise RMSD                 | 2.89         | 7.01   |
> | Pairwise RMSD r               | 0.48         | 0.08   |
> | RMSF                          | 1.68         | 3.74   |
> | Global RMSF r                 | 0.6          | 0.19   |
> | Per target RMSF r             | 0.85         | 0.67   |
> | RMWD                          | 2.61         | 7.27   |
> | RMWD trans                    | 2.28         | 5.22   |
> | RMWD var                      | 1.3          | 4.35   |
> | MD PCA W2                     | 1.52         | 2.06   |
> | Joint PCA W2                  | 2.25         | 5.97   |
> | PC sim > 0.5 %                | 44           | 21.95  |
> | Weak contacts J               | 0.62         | 0.45   |
> | Weak contacts nans            | 0.41         | 0      |
> | Transient contacts J          | 0.5          | 0.26   |
> | Transient contacts            | 0.25         | 0      |
> | Exposed residue J             | 0.50          | NaN    |
> | Exposed MI matrix rho         | 0.25          | NaN    |
>
> From the results above, it is evident that the ESM3 struggles with modeling the “local” and “detailed” structural changes in the 100ns simulation Atlas dataset, as many conformations may cluster within the xyz neighborhood. This highlights the advantages of explicitly generative modeling in Euclidean space for this Atlas task (their specific data distribution), where the AlphaFlow model can be more precisely defined here. Building on this observation, we propose that a promising direction for future research lies in developing novel SLM architectures tailored to MD data, since it is not trivial to directly apply to. These architectures should incorporate inductive biases that align with the local exploration characteristics of MD simulations, which could involve designing more effective auto-encoders and integrating advanced structure language model priors.
>
> (to be continued)

---

> > ### Author Response · Authors · 2024-11-17
> > **Response to reviewer yRU8 - Part 2**
> >
> > We want to further discuss the key difference between MD dataset and long dynamics of BPTI.
> > - Most of the structural proteins (such as in Atlas, MDCath) cannot experience large structural changes in a limited simulation time scale. For equilibrium sampling, Atlas and MDCath limit the time scale to ~100ns (100ns for Atlas and hundreds of ns for MDCath, depending on the targets). Even though they conducted simulations in atomic scale and used explicit water models, which is good in practice, the structure fluctuation/flexibility is quite limited. For example, if some model predicts a typical meta-stable conformation appearing in the 100us simulation or 1ms simulation, the distribution comparison will add a penalty to it.
> > - For BPTI trajectory, because it is a very classic and well accepted. We believe that the generative models that can infer the kinetic clusters without even seeing the true dynamics (following the sense in Lu et al. and Wang et al.). This reveals some commonsense shared across the general PDB database and physical force fields.
> > - As a probing investigation, we reported several trajectory RMSD in mean and max and saw a surprisingly low RMSD within each Atlas trajectory.
> >
> > | Test Target  | Trajectory RMSD-mean | Trajectory RMSD-max |
> > |-----------|--------------|--------|
> > | 6o2v_A | 1.345 A | 2.619 A |
> > | 7ead_A | 1.568 A | 2.296 A |
> > | 6uof_A | 0.803 A | 1.237 A |
> > | ...|...|...|
> > | All Test split (mean)|2.042 A| 3.572 A|
> >
> > From above we see the Atlas conformations generally have low RMSD to each other. Since Atlas targets are general monomeric proteins with hundreds of residues, the average trajectory RMSD ranging 1-2 angstrom is too small to reveal significant changes. That means the conformations are nearly never changed and will favor models with limited and local exploration such as AlphaFlow and currently not suitable for current ESM3 (as well as its latent space). In summary, this key observation inspires us to perform non-trivial modifications to make it work for the MD data.
> >
> > **quality** - (2) For iterative decoding, we note that the “iterative decoding” is exactly the “minibatch gibbs” sampling used across our study and explained in the original appendix (sorry for the confusion). Thanks very much for suggesting this and it leads us to make further remarks to indicate this. In other words, the pretrained ESM3 (zero-shot) is evaluated under the “iterative decoding” while in more exhaustive steps (N=25).
> >
> > **quality** - (3) For the result mixing,
> > We carefully examine the instance/metrics that underperform zero-shot and express our reasoning as follows: (1) our designed evaluation/test set covers a broad range of protein conformation behaviors (equilibrium dynamics, fold switchers, and IDP), (2) ESM is a large model training on the large dataset (across PDB, AFDB, ESMAtlas, etc.), that should cover a well diversified instances including IDP/IDR and probably the meta-stable structures. (3) the metrics that “zero-shot > ESMdiff” fall in the RMSD[“kinetic cluster”, table2] and MAE[“IDP test set”, table 4]. So the fall-short of ESMdiff maybe caused by the distribution shift during finetuning with a “biased (more structural)” training data. Since all our experiments are based on a training based on PDB monomer data in X-ray/Cyro-EM (similar to AF training set), we conceive that in the near future, one promising and straightforward way to improve the ESMDiff performance is to incorporate a diverse set of high-quality structure data including AFDB database, NMR data, etc.

---

> > > ### Author Response · Authors · 2024-11-17
> > > **Response to reviewer yRU8 - Part 3**
> > >
> > > For **clarity**,
> > >
> > > We again appreciate the valuable feedback from this point. We carefully read your suggestion and then address all of them according to the following points:
> > > - We have added additional “colored notation” to mark the best digit across all benchmarks tables v.s. among proposed SLM family.
> > > - We refine the relevant texts during the “probabilistic framework” for the confusing sentences there. To briefly answer here, (1) we did not further train the tokenizer; (2) that’s a limitation of their open-source version, because no side chain is predicted unlike their paper. For this, we add additional remarks there to better clarify.
> > > - The motivation of discuss the theoretical framework is to inspire and welcome interesting future work from the ML community. Researchers who are from the ML side would benefit from a familiar framework to work on this direction. Though this may be viewed as an "off-the-shelf" application/finetuning, we here want to note the SLM can be extended to other choices of tokenizers (encoder and decoder) based on optimizing the conditional ELBO.
> > > - For the general discrete diffusion, we agree with this opinion and move the lengthy paragraphs to appendix for reader’s self-contained reference. At the same time, we bring front the “inpainting case study of nanobody” to the main part, which we believe is more attractive from the application perspective.
> > > - For the difference between ESMDiff and ESM all-mask level training, we think this is a good and thoughtful question. Time-conditioning and stochastic unmasking are two of the factors,  but we care more about the “adaption purpose”. Okay, given the pretrained ESM, this is a very versatile foundation model that can do anything, since it theoretically covers all mask scales. However, it may be too “foundation” (broadly useful) to practically work in specific tasks especially for its open-source model with limited size (1.4B), for example, in conformation generation, we provide the whole sequence as the condition (partially for ESM3 training), and do not actually need function outputs (GO term). In our preliminary experiment, we have seen that mixing the noise / masking simultaneously from different modality tracks slows the training and even makes the learning more unstable, thus weakening the specificity for downstream tasks. We used to also conduct trials using the linear masking schedule similar to the ESM paper but it didn’t work well. Also, we believe one of our contributions is providing a clear recipe to easily (training, using specific noise schedule) adapt and bring it to many other realistic tasks.
> > >
> > > In summary, we reorganized the paragraphs and introduced remarks that help grasp/simplify the idea according to your suggestion above. We value any of your helpful comments!
> > >
> > >
> > >
> > > [1] Lin, Xiaohan, et al. "Tokenizing Foldable Protein Structures with Machine-Learned Artificial Amino-Acid Vocabulary." bioRxiv (2023): 2023-11.
> > >
> > > [2] Gao, Zhangyang, et al. "Foldtoken: Learning protein language via vector quantization and beyond." arXiv preprint arXiv:2403.09673 (2024).
> > >
> > > [3] Hayes, Tomas, et al. "Simulating 500 million years of evolution with a language model." bioRxiv (2024): 2024-07.

---

> > > > ### Author Response · Authors · 2024-11-25
> > > > **Response to reviewer yRU8**
> > > >
> > > > Dear reviewer yRU8,
> > > >
> > > > Thank you again for your time and effort in reviewing our manuscript : ) As the author-reviewer discussion deadline is approaching, we would like to follow up to see if the response addresses your concerns or if you have any further questions or recommendations, which we would highly appreciate!
> > > >
> > > > Author #5043

---

> > > > > ### Comment · Reviewer_yRU8 · 2024-11-26
> > > > >
> > > > > Thanks for the detailed response and additional experiments. I appreciate the significant rewrite of Section 3 and the addition of the nanobody results. I also appreciate the new ATLAS experiments; however, the authors seem to argue that because ATLAS is an imperfect dataset, that it disadvantages their method over other methods. I do not find this argument convincing, and think it is valuable to transparently show the performance of the method on as many axes as possible.
> > > > >
> > > > > With that said, I am happy enough with the to raise my score to a 6 or 7 if the authors do the following
> > > > > * Rename "minibatch Gibbs sampling" everywhere since it is not Gibbs sampling.
> > > > > * Add the ATLAS results to the appendix and at least reference it in the main text. (You can use ESM3 results now and replace with ESMDiff in the camera-ready)
> > > > > * Make even more clear (i.e., what you say in the rebuttal) the difference between ESMDiff and ESM3 in the main text

---

> ### Author Response · Authors · 2024-11-26
> **Response to Reviewer yRU8**
>
> Dear reviewer yRU8,
>
> Thanks very much for your follow-up suggestion. We apologize if our response implied any such intention. We did not mean to argue or justify whether Atlas is imperfect.
> Our prior response **"Atlas dataset is a bit special for their data distribution and not fully suitable for the current application of ESM-like model…"**  was to provide a possible explanation for why language models may fall short of Euclidean-space diffusion models (eg., AlphaFlow), especially in the 100ns MD data setting.
>
> Nevertheless, we believe all these mentioned datasets are valuable and cover different aspects, and thus we discuss them to provide more context in the rebuttal. Surely, for more comprehensive benchmarking purposes, we are happy to include the additional results in the manuscript, as you suggested; we also believe it may provide good context to inspire valuable future works!
>
> To conclude, we have addressed these in the updated pdf:
> - (Globally) Rename "minibatch Gibbs sampling" -> **I**terative **D**ecoding  (ID), according to the naming convention in ESM3 preprint paper (A.1.10 - ESM3 Inference, Hayes et al. [1])
> - Added the MD results from rebuttal to Appendix E.4/Table S6 (+will complement the table and enrich the discussion in the final version)
> - Incorporated the difference between ESM3 and ESMDiff in Appendix B.4 and also put a reference anchor in the first paragraph of Section 5.
>
> We want to extend our gratitude again for your helpful assessment and supportive comments for improving the quality of our manuscript, as well as pointing out the promising follow-up directions for future research :)
>
> [1] Hayes, Tomas, et al. "Simulating 500 million years of evolution with a language model." bioRxiv (2024): 2024-07.

---

> > ### Comment · Reviewer_yRU8 · 2024-11-26
> >
> > Thanks for the changes. The formatting and presentation of the ATLAS results will need some work; however I appreciate the authors accommodating them in spirit and will raise my score to 6. Also don't forget to change the "Gibbs" labels in all the figures!

---

> > > ### Author Response · Authors · 2024-11-27
> > > **Response to Reviewer yRU8**
> > >
> > > Thanks for the kind reminder of the figure label :) We have had them accordingly corrected! Thank you!

---

### Official Review · Reviewer_FWmx · 2024-11-04

**Soundness:** 3
**Presentation:** 3
**Contribution:** 3
**Rating:** 8
**Confidence:** 4

**Summary:**

In this paper, the authors present a novel Structure Language Modeling (SLM) framework that integrates recent advances in discrete variational autoencoders (dVAEs) to quantise the latent space and language models to learn the conditional distribution on this latent space with masked diffusion models framework. This combined approach is applied to model the conformation space of proteins. The proposed framework offers flexibility in selecting the dVAE and language model components; however, the authors introduce a specific setup, named ESMDiff, which utilizes ESM3 and BERT as its primary components. To demonstrate the effectiveness of this approach, the authors evaluate ESMDiff on several case studies, including the structural dynamics of bovine pancreatic trypsin inhibitor (BPTI), modeling conformational changes in structural proteins, and exploring the conformation space of intrinsically disordered proteins (IDPs).

**Strengths:**

The paper is well-structured, with a clear and comprehensive presentation of the theoretical background and framework. Although flexible learnable priors in VAEs have a substantial history of research, the idea of coupling the dVAE and language models as flexible conditional prior is fresh and novel in the area of protein conformation modeling. The selected case studies are relevant and show the potential for wide practical application.

**Weaknesses:**

While the overall quality of the paper is high, there are a few weaknesses and areas that could benefit from further clarification. Details on these points are provided in the Questions section.

**Questions:**

* [major] While the experiments compare a diverse set of baseline models, they lack a comparison with pure language models. Previous work [a] demonstrated that language models can generate molecular and protein structures with atomic coordinates represented as regular text, and other research [b] has shown that this approach is viable for molecular conformation space modeling. Given the context sizes of modern language models, it seems feasible to model protein backbone coordinates directly as text. Including a comparison with pure language models could provide a strong justification for using dVAE in this framework.
* [minor] In reference to lines 175–186, the training of the dVAE and the language model prior happens in two separate stages. Previous research on learnable priors in VAEs [c, d] showed that the encoder, decoder, and prior can be trained simultaneously. It would be helpful to clarify, either theoretically or through an ablation study, why this work uses a multi-stage training approach rather than a joint training method, as this separation introduces additional complexity to the training process.
* [minor] The scope of this work is limited to modeling the coordinates of the protein backbone (lines 145–150). However, accurate spatial positioning of side chains is essential for many practical applications, especially when modeling interactions between the protein and other proteins or molecular structures is required [e]. Including a discussion of this limitation in the Limitations section would strengthen the paper by addressing the potential impact of excluding side chains on the model’s practical applications.

a. Language models can generate molecules, materials, and protein binding sites directly in three dimensions as XYZ, CIF, and PDB files, 2023

b. BindGPT: A Scalable Framework for 3D Molecular Design via Language Modeling and Reinforcement Learning, 2024

c. VAE with a VampPrior, 2017

d. A Prior of a Googol Gaussians: a Tensor Ring Induced Prior for Generative Models, 2019

e. Rotamer Density Estimator is an Unsupervised Learner of the Effect of Mutations on Protein-Protein Interaction, 2023

---

> ### Author Response · Authors · 2024-11-17
> **Response to Question 1 from Reviewer FWmx**
>
> Dear reviewer FWmx,
>
> We want to extend our heartfelt thanks for your supportive comments on our manuscript and the detailed suggestions. Here is the response to Question 1[major]:
>
> Thank you so much for suggesting the related work. We agree that pure LM with structure coordinates as text is a fairly parallel work to our method given similar data and model architecture. Yet neither of the papers has released their code. The official repos of both LM-CH [a] and BindGPT [b] [https://github.com/danielflamshep/xyztransformer, https://github.com/insilicomedicine/bindgpt ] are yet empty :( We would like to add the comparison once the code is available making our experimental benchmark more comprehensive.
>
> That being said, we add the following discussion regarding the pure LM over coordinates and explain why our approach may be better in the current revision:
> - Firstly, formulating the conformation generation problem as pure “text” generation and using atomic coordinates directly as the target requires the LM to handle longer context dependencies due to the tokenization manner, which adds additional challenges to the learning problem[1,2].
> - Also, taking [b] for example, the total number of atoms in the “small molecules” is limited only to tens to hundreds, and thus learning on the system may not suffer from long context or scaling. However, without proper coarse-graining, successful models on small molecules may not work as well for large molecules such as proteins of thousands of atoms. How to properly coarse-grain the xyz coordinate file is a non-trivial problem in practice.
> - Moreover, without further modifications, cross-entropy loss on the text tokens of xyz digits does not serve as a good measure for the distance between geometric structures and thus the space structure to be learned. After the training, the GPT model may achieve a high likelihood by only capturing high-frequency xyz patterns instead of the low-frequency features that are critical for determining valid structures. For similar reasons, the tasks that use LMs to generate continuous data, such as text-to-image and video translation, often involve the LM predicting quantized features produced by a VQ-VAE [3,4,5].
> - Last but not least, the conformation generation requires some configuration information about the protein structure we want to model (in our case, the amino acid types). This is however different from these models which aim to de novo “design” unnatural molecules from scratch. How to place such conditions in these models is still a good open question.
>
> We value this interesting open discussion, and we think it is important to design and encode proper inductive bias into modeling the xyz coordinate in the file data and make these LM models more promising in parallel to our SLM!
>
> [1] Kuratov, Yuri, et al. "BABILong: Testing the Limits of LLMs with Long Context Reasoning-in-a-Haystack." arXiv preprint arXiv:2406.10149 (2024).
>
> [2] Lee, Jinhyuk, et al. "Can Long-Context Language Models Subsume Retrieval, RAG, SQL, and More?." arXiv preprint arXiv:2406.13121 (2024).
>
> [3] Van Den Oord, Aaron, and Oriol Vinyals. "Neural discrete representation learning." Advances in neural information processing systems 30 (2017).
>
> [4] Yan, Wilson, et al. "Videogpt: Video generation using vq-vae and transformers." arXiv preprint arXiv:2104.10157 (2021).
>
> [5] Bordes, Florian, et al. "An introduction to vision-language modeling." arXiv preprint arXiv:2405.17247 (2024).

---

> ### Author Response · Authors · 2024-11-17
> **Response to Question [2,3] from Reviewer FWmx**
>
> **Response to Q2:**
>
> Thanks for this point. We believe that joint training for the discrete VAE and LM prior is not very feasible in this context, as it requires backpropagation through latent tokens to LM forward on the fly. We attempt to answer this question from the theoretical perspective (we apologize in advance if we misunderstood the question.):
> - Joint training of LM prior and the dVAE may suffer from unstable training. During the early training stage, the dVAE posterior can generate sequences of latent tokens that may be out-of-distribution along with the training. This is because the posterior p(z|x,c) is parameterized by a categorical distribution over latent tokens such that it can more drastically change when the model parameters in p(z|x,c) are updated along with training. This can confuse the LM prior and slow down the training.
> - Secondly, we notice that previous practices in the image, video, etcs. [1,2,3] usually choose to place the training for the autoencoder (constructing the latent space) before the prior training. This makes sense because the structure of the latent space is definite and fixed during the training of LM prior, guaranteeing that the training can converge in a stable way. A very long-standing example in joint training of generative models is the GAN[4], which has strong performance yet suffers from unstable training and tricky convergence.
>
> We agree that the end-to-end joint training seems more attractive yet it may practically be difficult for training due to the interplay between different components, in specific the hyperparameter tuning of loss weights, learning schedule, etc. We are with the idea that training separately may result in suboptimal loss minima yet it can converge better in practice, and thus we follow prior works like [1,2,3] to derive the scheme. We kindly hope this can help explain the situation better!
>
> [1] Van Den Oord, Aaron, and Oriol Vinyals. "Neural discrete representation learning." Advances in neural information processing systems 30 (2017).
> [2] Ramesh, Aditya, et al. "Zero-shot text-to-image generation." International conference on machine learning. Pmlr, 2021.
> [3] Yan, Wilson, et al. "Videogpt: Video generation using vq-vae and transformers." arXiv preprint arXiv:2104.10157 (2021).
> [4] Goodfellow, Ian, et al. "Generative adversarial nets." Advances in neural information processing systems 27 (2014).
>
>
> **Response to Q3:**
>
> - We sincerely appreciate this biologically relevant suggestion. We agree with the idea that building an atom-level autoencoder including the side chains, and even other biomolecules such as ligands, RNA, etc. will make the generative model much more powerful in downstream applications. We have briefly mentioned this in the **Section 6 (Conclusion and limitations)** to potentially enlighten valuable future works!

---

> > ### Comment · Reviewer_FWmx · 2024-11-21
> >
> > I sincerely thank the authors for providing comprehensive responses to my questions and suggestions. I remain confident that the paper is well-written and the results are justified, and the research direction is promising. I will therefore maintain my score.

---

> > > ### Author Response · Authors · 2024-11-22
> > > **Response to Reviewer FWmx**
> > >
> > > Dear Reviewer FWmx,
> > >
> > > Thanks very much for your support and helpful suggestion. The discussion did inspire us in many ways and also point out promising future directions, which we highly appreciate :)
> > >
> > > Authors #5043

---

### Meta-Review · Area_Chair_1wCJ · 2024-12-17

**Metareview:**

The submission presents a protein conformation generative model based on the VAE framework, where the latent space is discrete and sequential, on which the prior is flexibly modeled by a language model that can properly handle a given protein sequence. The encoder and decoder have properly taken geometric invariance into consideration. Although using a language model, the proposed formulation can still capture the spatial interactions mutually between every pair of residues, and the discretization does not lose the spatial resolution.

I agree with the reviewers' appreciation on the soundness, good presentation, and the promising future potential. Reviewers also pointed out a few insufficiencies, including lack of comparison with a pure language model, limited evaluation cases, and some mismatches between the introduction of a general framework and the actually employed method (I don't think the former is unnecessary, though). In the rebuttal, the authors updated the presentation, and presented some results on nanobody and ATLAS, which have addressed some of the limitations, while some remains (e.g. limited advantageous results on ATLAS, alignment and presentation in metric, model, and dataset settings for comparisons with other methods). Overall, I agree with the reviewers' opinion to appreciate the contributions over the remaining limitations, hence recommend an accept, and hope the authors improve the manuscript further according to the feedback.

**Additional Comments On Reviewer Discussion:**

Reviewers yRU8 and X2Sg gave some critical feedback on the original draft, including presentation clarity (alignment of the general method and the practically used version), comparison with the pure language model counterpart ESM3, alignment of dataset and metrics for comparison. In the rebuttal, the authors have modified the paper for better methodology presentation, provided preliminary comparison results with ESM3, and presented comparison results in a simplified but aligned setting and explained difficulties for further comparison. The reviewers found the updates sufficient to cover the major concerns and increased their scores to positive accordingly. I hold the same opinion that although not perfect, the results support promising evidence given reasonable effort, and the contributions could outweigh the remaining limitations overall.

---

### Decision · Program_Chairs · 2025-01-22

Accept (Poster)